# CoRGI 🐱: GNNs with Convolutional Residual Global Interactions for Lagrangian Simulation

## Abstract

Partial differential equations (PDEs) are central to dynamical systems modeling, particularly in hydrodynamics, where traditional solvers often struggle with non-linearity and computational cost. Lagrangian neural surrogates such as GNS and SEGNN have emerged as strong alternatives by learning from particle-based simulations. However, these models typically operate with limited receptive fields, making them inaccurate for capturing the inherently global interactions in fluid flows. Motivated by this observation, we introduce **Co**nvolutional **R**esidual **G**lobal **I**nteractions (CoRGI), a hybrid architecture that augments any GNN-based solver with a lightweight Eulerian component for global context aggregation. By projecting particle features onto a grid, applying convolutional updates, and mapping them back to the particle domain, CoRGI captures long-range dependencies without significant overhead. When applied to a GNS backbone, CoRGI achieves a 57% improvement in rollout accuracy with only 13% more inference time and 31% more training time. Compared to SEGNN, CoRGI improves accuracy by 49% while reducing inference time by 48% and training time by 30%. Even under identical runtime constraints, CoRGI outperforms GNS by 47% on average, highlighting its versatility and performance on varied compute budgets.

## 1 Introduction

### 1.1 PDEs and Discretization

Partial differential equations (PDEs) provide the governing framework for transport, diffusion, and wave phenomena in continua (e.g. Euler (1757); Fourier (1822); Maxwell (1873)). In fluid mechanics, the conservation of mass, momentum, and energy is expressed by the Navier–Stokes equations. Because closed-form solutions are rarely available for realistic geometries or parameter regimes, computational fluid dynamics (CFD) employs numerical discretizations to approximate these PDEs.

A central modeling choice concerns the kinematic frame, of which two are prevalent. In an **Eulerian** formulation (e.g. (LeVeque, 2002; 2007; Zienkiewicz et al., 2005)), the unknown fields are represented on control volumes fixed in physical space. Conservation laws are enforced through fluxes across the faces of these volumes. In a **Lagrangian** formulation, the mesh points or particles move with the material velocity. Advection becomes a time derivative along trajectories. For CFD, it is common to use SPH, which computes spatial derivatives through local interpolation among neighboring particles (Monaghan, 2005; Price, 2012). However, these kinematic frames are not mutually exclusive: particle–mesh hybrids such as PIC (Evans & Harlow, 1957), FLIP (Brackbill & Ruppel, 1986), and MPM (Sulsky et al., 1994) advect mass or momentum on particles but solve pressure or diffusion on a background grid. As Eulerian methods tend to be weaker at modeling particle-particle interactions and Lagrangian methods tend to struggle on nonlocal constraints, CoRGI is motivated by combining the strengths of both approaches.

### 1.2 Machine Learning Surrogates for PDEs

In recent years, the rise of artificial intelligence in scientific discovery has opened new pathways for solving PDEs using neural PDE surrogates, promising faster and more scalable simulations

by learning to approximate PDE solutions. Notable approaches include Physics-Informed Neural Networks (PINNs) (Raissi et al., 2019), Fourier neural operators (FNOs) (Li et al., 2020), and DeepONets (Lu et al., 2021), which learn mappings between function spaces instead of pointwise solutions. However, most of these efforts target **Eulerian** representations, where the input structure lends itself naturally to convolutional architectures and structured grids.

Modeling **Lagrangian** systems requires handling unstructured particle clouds and irregular interactions. Graph neural networks (GNNs) have emerged as a natural fit for this setting, offering the flexibility to learn local interactions between particles (Battaglia et al., 2016; Sanchez-Gonzalez et al., 2020). LagrangeBench (Toshev et al., 2023) introduced a benchmark suite for evaluating GNN-based surrogates on various fluid systems, including dam breaks, lid-driven cavities, and Taylor-Green vortices, providing a much-needed standardization in this space. Despite their expressiveness, GNNs are computationally limited: long-range interactions require deep GNN stacks, which are prone to over-smoothing and memory inefficiency.

### 1.3 CORGI: A HYBRID ARCHITECTURE

Traditional solvers have long recognized the strengths of combining Eulerian and Lagrangian discretizations. While Lagrangian methods excel at local accuracy and topological flexibility, Eulerian grids offer efficient global operations and regular data structures. In this work, we pursue a neural hybrid approach: **an Eulerian augmentation of Lagrangian GNNs**. Our key observation is that local feature aggregation — well handled by GNNs — scales well in large systems due to the spatial locality of interactions, while the bottleneck of global aggregation can be addressed through grid-based projection of particle features. In particular, our intuition is that emulation of SPH may fail to capture global dynamics (e.g. inverse energy cascades) and that emulation of MPM may fail to capture local dynamics (i.e. particle-particle interactions), but our method synthesizes both techniques without being computationally prohibitive. To our knowledge, such a synthesis remains underexplored in the traditional methods literature, and as a result, is similarly untapped within the neural surrogates literature.

We introduce **Co**nvolutional **R**esidual **G**lobal **I**nteractions (CORGI), a CNN-driven graph neural network variant that enhances their ability to model long-range interactions in Lagrangian PDE systems. CORGI projects intermediate graph features onto an Eulerian grid, performs efficient global aggregation using convolutions, and maps the result back to the particle domain for downstream processing. This design retains the local modeling power of GNNs while offloading global feature handling to CNNs, resulting in a hybrid surrogate that is both expressive and efficient.

Our main contributions are as follows:

- We identify a computational bottleneck in GNN-based Lagrangian surrogates arising from long-range interactions, and addresses it by proposing a novel hybrid architecture CORGI that augments GNNs with CNN-based global aggregation via a learned Eulerian transformation.

- We validate CORGI through extensive experiments on the LagrangeBench suite, demonstrating that CORGI on average doubles rollout prediction accuracy across all benchmark tasks, with 13% runtime overhead.

- We further demonstrate CORGI enables shallower, faster GNNs without performance decrease: CORGI performs on average 47% better than GNS with the same time budget.

### 1.4 RELATED WORKS

For our model architecture, we conducted a survey of the literature to determine both our Lagrangian and our Eulerian components.

**Eulerian surrogates.** A U-Net (Ronneberger et al., 2015) based Eulerian surrogate, compared to many related methods (e.g. FNO (Li et al., 2021), F-FNO (Tran et al., 2021), GINO (Li et al., 2023), transformer-based neural operators (Cao, 2021; Hao et al., 2023; 2024; Li et al., 2022; Wang et al., 2024)), was found to be more efficient for a given input resolution during inference time by Alkin et al. (2025). Other Eulerian methods present other challenges: MeshGraphNets (Pfaff et al., 2021) are known to struggle with non-local features; its extension to multiscale (Fortunato et al., 2022) requires the generation of additional meshes. We utilize a grid for this coarser mesh, which is more amenable

to efficient convolution operations than other types of meshes. We use a grid-based convolution instead of a continuous convolution (Ummenhofer et al., 2019) as they were found to perform better at a lower computational cost by Rochman-Sharabi et al. (2025). Nonetheless, some Eulerian or hybrid methods are comparable to ours, and hence we provide comparisons to UPT (Alkin et al., 2025) and NeuralMPM (Rochman-Sharabi et al., 2025) in App E.

As discussed in App F, using efficient adaptive mesh refinement while preserving high efficiency remains an open challenge. We leave the integration of the more costly surrogates into the CORGI framework for future work.

**Lagrangian surrogates.** We identify two architectures considered state of the art for Lagrangian simulations: GNS (Sanchez-Gonzalez et al., 2020) and SEGNN (Brandstetter et al., 2022). GNS is a neural simulator which learns graph message passing in order to predict dynamics from a graph of particles; we elaborate upon this in Sec 2.4. SEGNN is another simulator which enforces equivariance in its update and message passing functions and utilizes steerable feature vectors for representing covariant information. GNS (Sanchez-Gonzalez et al., 2020) is considered the fastest Lagrangian surrogate due to its simplicity (Toshev et al., 2023), and hence serves as our backbone in the main text. Many popular architectures (Anderson et al., 2019; Batzner et al., 2022; Brandstetter et al., 2022; Thomas et al., 2018) rely on expensive Clebsch-Gordan tensor products, but we nonetheless also provide comparisons (Table 1) to SEGNN (Brandstetter et al., 2022), demonstrating its inefficiency compared to CORGI.

We observe that these models only facilitate local interactions. We note that a failure mode of these models is their poor enforcement of fluid incompressibility (Fig 2). We hypothesize this is due to the low propagation speed of purely local GNNs, which we address in CORGI by increasing the receptive field of each node, thus increasing the propagation speed of our architecture without increasing temporal resolution. Given the improved accuracy using CORGI compared to the other architectures, we identify CORGI as Pareto optimal.

To our knowledge, the most similar contribution in the literature to CORGI is Neural SPH (Toshev et al., 2024), which introduces spherical particle hydrodynamics relaxation steps during inference time. However, Neural SPH does not change the neural model itself, only providing an inference time adjustment. In addition, this adjustment relies on normalization statistics, making it brittle in unknown scenarios. Neural SPH and CORGI are orthogonal augmentations to Lagrangian neural networks: Neural SPH applies a local correction, while CORGI aims to capture global features. As we improve different aspects of neural surrogates, these approaches are complementary, i.e. it is possible to apply both CORGI and Neural SPH to GNNs.

## 2 PRELIMINARIES

**Notations.** In the text, we utilize Hadamard notation frequently in representing our operations, similar to what one might expect in various computer programming contexts. We generally use regular lower case letters to denote scalars, bold lower case letters to denote vectors, and bold upper case letters to denote matrices and high-dimensional tensors. We represent the concatenation of features $\mathbf{x}$ and $\mathbf{y}$ as $\mathbf{x} \oplus \mathbf{y}$, and the composition of $f$ and $g$ as $f \circ g$. For integers $a < b$, we use $[\![a, b]\!]$ to denote the set of integers $\{a, a+1, \cdots, b\}$. Our variables include $i, j \in [\![1, N]\!]$ to enumerate particles, $\ell$ to enumerate message passing steps, and $k$ to represent the number of levels for convolution. All other variables are introduced as they appear in the text.

### 2.1 COMPUTATIONAL FLUID DYNAMICS (CFD)

We specifically focus on applying our method to CFD simulations, which are governed by the Navier-Stokes equation. Unlike the heat or wave equation, the Navier-Stokes equation is a nonlinear PDE and is more challenging to simulate. The general form of the equation is

$$\rho \left( \frac{\partial}{\partial t} + \mathbf{u} \cdot \nabla \right) \mathbf{u} = -\nabla p + \nabla \cdot \boldsymbol{\tau} + \rho \mathbf{a}$$

where $\rho$ is the density, $\mathbf{u}$ is the flow velocity, $p$ is the pressure, $t$ is time, $\boldsymbol{\tau}$ is the Cauchy stress tensor, and $\mathbf{a}$ represents externally induced acceleration, e.g. gravity. In the incompressible regime, $\boldsymbol{\tau} = \mu \left[ \nabla \mathbf{u} + (\nabla \mathbf{u})^{\mathsf{T}} \right]$, where $\mu$ is the dynamic viscosity; we note that incompressibility implies

being divergence free, i.e. $\nabla \cdot \mathbf{u} = 0$. The quadratic advective term $(\mathbf{u} \cdot \nabla)\mathbf{u}$ invites both local and global scale phenomena; as it is nonlinear, this prevents solutions for different parts of the flow from being independent of one another: for instance, in 2D flows, it is common to observe inverse energy cascades, i.e. energy is transferred from smaller scales to larger scales. We wish to incorporate this mathematical prior by utilizing a neural surrogate for both local and global modes; methods like SPH by themselves struggle with larger scales, as the diffusion and pressure projection steps require large neighborhoods to be accurate.

## 2.2 PROBLEM STATEMENT

Our aim is to autoregressively estimate future positions of particles $\hat{\mathbf{x}} \in \mathbb{R}^{N \times d}$. To do so, we define a graph network surrogate that predicts their acceleration, i.e. $\mathcal{F}_\theta : (\mathcal{G}_0, \mathbf{x}_0, \mathbf{p}) \mapsto \hat{\ddot{\mathbf{x}}} \in \mathbb{R}^{N \times d}$, where $N \in \mathbb{N}$ is the number of particles, $d \in \{2, 3\}$ is the spatial dimension, $\mathbf{x}_0 := (x_i : i \in [\![1, N]\!]) \in \mathbb{R}^{N \times d}$ is the current particle positions, $\mathcal{G}_0 = (\mathcal{V}, \mathcal{E})$ is a graph (such that $\mathcal{V}$ represents particles and $\mathcal{E}$ is constructed based on distance) where $|\mathcal{V}| = N$, $\mathbf{p} \in [\![1, T]\!]^N$ is the particle types, and $\hat{\ddot{\mathbf{x}}}$ is the predicted acceleration. Then, positions at the following time-step are estimated by taking the symplectic Euler integral (Vogelaere, 1900): $\hat{\dot{\mathbf{x}}}^{(t+1)} = \hat{\dot{\mathbf{x}}}^{(t)} + \Delta t \hat{\ddot{\mathbf{x}}}^{(t)}; \hat{\mathbf{x}}^{(t+1)} = \hat{\mathbf{x}}^{(t)} + \Delta t \hat{\dot{\mathbf{x}}}^{(t+1)}$.

## 2.3 GRAPH ENCODING

The multilayer perceptrons (MLP) (Rumelhart et al., 1986) $\varphi$ used below are the standard implementation with ReLU activation (Agarap, 2019) and LayerNorm normalization (Ba et al., 2016), hence their mathematical formulations are not provided. Similarly, we use the standard convolution (Lecun et al., 1998) and transposed convolution (Dumoulin & Visin, 2016) layers. We use a stride of 2 for pooling operations unless stated otherwise.

We use a graph to facilitate particle-particle/local interactions. We consider the graph at time $t$ denoted by $\mathcal{G} = (\mathcal{V}, \mathcal{E}, \{\boldsymbol{h}_i^0\}_{i \in \mathcal{V}}, \{\boldsymbol{e}_{ij}^0\}_{(i,j) \in \mathcal{E}})$, where i.e. an object with $\mathcal{V}$ denotes vertices, $\mathcal{E}$ denotes edges, and $\boldsymbol{h}_i^0, \boldsymbol{e}_{ij}^0$ denote node and edge features, respectively. We use initial node and edge features

$$\boldsymbol{h}_i^0 = \left(\mathbf{x}_i^{t-\mathcal{L}:t} \oplus \tilde{\dot{\mathbf{x}}}_i^{t-\mathcal{L}:t-1} \oplus \|\dot{\mathbf{x}}_i^{t-\mathcal{L}:t-1}\| \oplus b_i \oplus \boldsymbol{f}_i \oplus \boldsymbol{t}_{p_i}\right) \in \mathbb{R}^{F_n} \tag{1}$$

$$\boldsymbol{e}_{ij}^0 = \left((\mathbf{x}_i^t - \mathbf{x}_j^t) \oplus \|\mathbf{x}_i^t - \mathbf{x}_j^t\|\right) \in \mathbb{R}^{F_e} \tag{2}$$

where $\mathcal{L} \in \mathbb{N}$ is the length of history we pass to the model, $\mathbf{x}_i^{t-\mathcal{L}:t} \in \mathbb{R}^{d \times \mathcal{L}}$ is the position history, $\tilde{\dot{\mathbf{x}}}_i^{t-\mathcal{L}:t-1} \in \mathbb{R}^{d \times (\mathcal{L}-1)}$ is the direction history (expressed as unit vectors), $\|\dot{\mathbf{x}}_i^{t-\mathcal{L}:t-1}\| \in \mathbb{R}^{d \times (\mathcal{L}-1)}$ is the speed history, $b_i \in \mathbb{R}$ is the distance from a boundary, $\boldsymbol{f}_i \in \mathbb{R}^d$ is the external forces vector, and $\boldsymbol{t}_{p_i} \in \mathbb{R}^{F_t}$ is the learned type embedding. For datasets with only one particle type, we omit $\boldsymbol{t}_{p_i}$ from the node features.

To complete the encoding, two MLPs, $\varphi_n : \mathbb{R}^{F_n} \to \mathbb{R}^H$ and $\varphi_e : \mathbb{R}^{F_e} \to \mathbb{R}^H$, yield latent embeddings $\boldsymbol{h}_i = \varphi_n(\boldsymbol{h}_i^0)$, $\boldsymbol{e}_{ij} = \varphi_e(\boldsymbol{e}_{ij}^0)$.

## 2.4 GRAPH NETWORK-BASED SIMULATORS (GNS)

The GNS is a graph-based Lagrangian surrogate which learns message passing for predicting dynamics introduced by Sanchez-Gonzalez et al. (2020). For each layer of the GNS, the latent graph is updated by 2 MLP layers with residual connections. Formally, for $\ell \in [\![1, L]\!]$, where $L \in \mathbb{N}$ is the number of message passing steps, $\mathcal{G}^{(\ell)} = \Psi^{(\ell)}(\mathcal{G}^{(\ell-1)})$,

$$\Psi^{(\ell)} := \left(1 + \varphi_n^{(\ell)} \circ \mathcal{U}_n\right) \circ \left(1 + \varphi_e^{(\ell)} \circ \mathcal{U}_e\right) \tag{3}$$

where $\varphi_e^{(\ell)} : \mathbb{R}^{3H} \to \mathbb{R}^H$ and $\varphi_n^{(\ell)} : \mathbb{R}^{2H} \to \mathbb{R}^H$ are both MLPs, $\mathcal{U}_e$ is the edge update map $\{\boldsymbol{h}_i, \boldsymbol{h}_j, \boldsymbol{e}_{ij}\} \mapsto \boldsymbol{h}_i \oplus \boldsymbol{h}_j \oplus \boldsymbol{e}_{ij}$, and $\mathcal{U}_n$ is the node update map $\{\boldsymbol{h}_i\} \cup \{\boldsymbol{e}_{ij}\}_{j:(i,j) \in \mathcal{E}} \mapsto \boldsymbol{h}_i \oplus \sum_{j:(i,j) \in \mathcal{E}} \boldsymbol{e}_{ij}$. This is the implementation found in Toshev et al. (2023), and hence the implementation used for our baseline and our augmentation. We provide the mathematical formulation here, as it slightly differs from the original formulation from Sanchez-Gonzalez et al. (2020). However, GNS models only the local interactions explicitly.

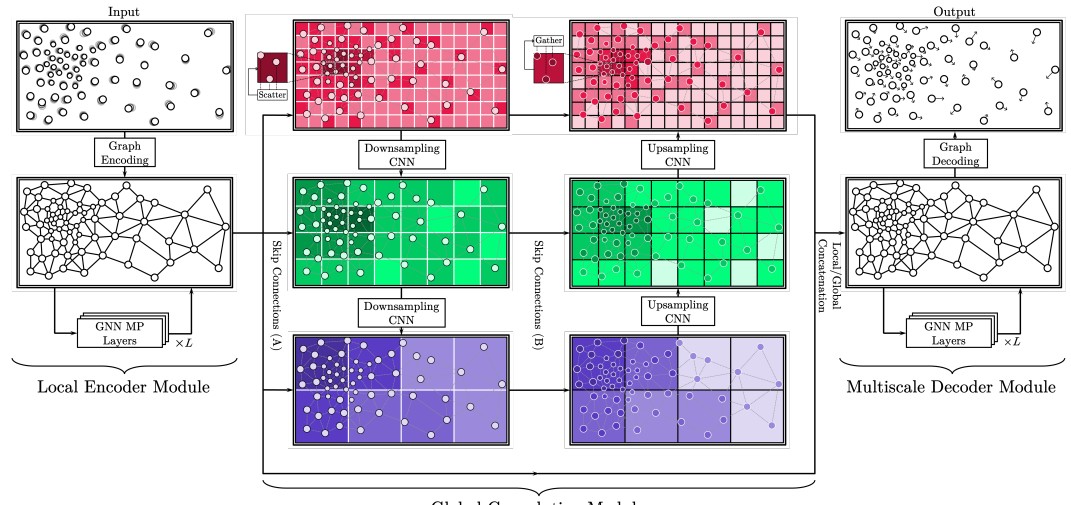

Figure 1: An illustration of the CORGI architecture over an example 2D particle dataset. For a GNN with $2L$ layers, our method takes node features encoded by the GNN at layer $L$, processes them into grid-based features at different resolution levels through a scatter operation, and then apply a multi-resolution CNN inspired by U-net (Ronneberger et al., 2015) to capture global features. After this, the processed grid feature at the highest resolution are mapped back to node features, and finally the latter $L$ layers of GNN message passing decodes the features into outputs.

## 3 METHODOLOGY

We observe that hydrodynamics often exhibits multiscale phenomena, and hence we wish to model nonlocal interactions. In particular, advection facilitates acoustically propagated local interactions, which are often approximated as nonlocal interactions as discussed in Sec 4.4. Unfortunately, increasing the interaction range scales computational costs by $\mathcal{O}(r^d)$. Hence, the contribution of CORGI is to incorporate global computation and information into Lagrangian surrogates via integrating with the Eulerian representation. We first generate representations utilizing a GNN encoder, then we perform convolution to expand a particle's receptive field, and lastly we fuse both the local and global information and utilize a GNN decoder. We describe in more detail the pipeline of CORGI in Fig 1.

### 3.1 FULL ARCHITECTURE OF CORGI

**GNN-based Local Encoder for Particles.** We utilize the architecture outlined in Sec 2.4 as an encoder with $L$ layers. Intuitively, it is desirable to have some topological and local geometric priors before we perform convolution. We denote its output $\boldsymbol{h}_i^{\downarrow}$.

**Grid Encoding via Scattering.** To assign values to our lattice, we use cloud in cell scattering (Birdsall & Fuss, 1969) on $\boldsymbol{h}_i^{\downarrow}$ to create a grid (referred to later as $G_0^{(1)}$). A precise formulation of this may be found in App E.1.4.

**Convolutional Global Interaction.** We use a convolutional component inspired by U-Net (Ronneberger et al., 2015) to allow for global feature interactions across long distances.

Given a tuple of CNN widths $(F_1, \ldots, F_K)$, we apply a convolution block $K$ times for downsampling:

$$G^{(k)} = \text{ConvBlock}_{F_k}\left(G^{(k-1)} \oplus G_0^{(k-1)}\right) \tag{4}$$

where $G_0^{(k)} = \text{AvgPool}\left(G_0^{(k-1)}\right)$ and $G^{(1)} = G_0^{(1)}$ is the output from the last step. For upsampling, we take $\tilde{G}^{(k-1)} = \text{ConvBlock}_{F_k}\left(\text{UpConv}\left(\tilde{G}^{(k)} \oplus G^{(k-1)}\right)\right)$. The output is $\tilde{G} = \tilde{G}^{(1)} \in \mathbb{R}^{|\mathcal{C}| \times H}$.

The convolution blocks above are standard for U-Net: double convolution using ReLU activation and InstanceNorm normalization (Ulyanov et al., 2017).

**Conversion to Particles via Gathering.** We use cloud in cell gathering on $\tilde{G}$, yielding particle vectors $\tilde{\boldsymbol{h}}_i$. We then augment the particle representations $\boldsymbol{h}_i^{\mathrm{aug}} = \boldsymbol{h}_i^{\downarrow} \oplus \tilde{\boldsymbol{h}}_i$.

**Multiscale Decoder Module.** We apply a linear projection $P \in \mathbb{R}^{2H \times H}$ and then reuse the GNS architecture from Sec 2.4 with $L$ layers using an independent set of parameters. In essence, we fuse global and local features for each particle for reconsideration in the local scale, which greatly improves the receptive field of each particle without significant compute overhead. We denote its output $\boldsymbol{h}_i^{\uparrow}$. We then use an MLP $\varphi_{\mathrm{dec}} : \mathbb{R}^H \to \mathbb{R}^d$ to yield an acceleration prediction $\hat{\ddot{\boldsymbol{x}}}_i = \varphi_{\mathrm{dec}}(\boldsymbol{h}_i^{\uparrow})$ for $i \in [\![1, N]\!]$.

## 3.2 ANALYSIS OF INFORMATION PROPAGATION

We further justify the importance of modeling dynamics globally, by considering the fact that advection propagates at the speed of sound. For a graph neural network whose neighborhoods are defined by a radius $r \in \mathbb{R}$, the maximum distance by which information can propagate for each time step is $r \times L$, where $L$ is the number of layers, therefore the corresponding propagation speed is upper bounded by $rL/\Delta_t$.

In order to increase propagation speed while keeping the GNN structure, one must either increase neighborhood radius, temporal fidelity, or number of message passing steps. However, all of these methods have major limitations: increasing the neighborhood radius inflates graph connectivity and hence memory costs by $\mathcal{O}(R^2)$; increasing temporal fidelity extends inference time by $\mathcal{O}(\frac{1}{\Delta_t})$, and may cause rollout instability; increasing the number of message passing steps also extends inference/training time by $\mathcal{O}(L)$, and may induce over-smoothing. CORGI, on the other hand, bridges this gap efficiently by directly learning global dynamics, relying on a separate Eulerian projection. We characterize the time complexity of CORGI with the following theorem.

**Theorem 1.** *Given GNN edge construction radius $r$, number of MP layers $L$, unit time step $\Delta_t$ and advection propagation speed $v$, the CNN module in CORGI require a minimum of $\Omega(\log(v\Delta_t/rL))$ layers to properly model long-range global interactions.*

We leave the proof of this theorem to App A. Note that as the number of particles $N$ increases, assuming we fix the Lebesgue measure of the ambient space, from the runtime and memory bottlenecks of GNN, $r$ must scale with $\mathcal{O}(1/N^{1/d})$, which means the inference and training time of the CORGI module scales roughly with $\mathcal{O}(\log N)$. This is a significant improvement compared with GNNs expanded to propagate information at scale with $v$, which suffers a complexity of $\mathcal{O}(L) = \mathcal{O}(1/r) = \mathcal{O}(N^{1/d})$.

## 4 EXPERIMENTS

In our experiments, we use the datasets, evaluation metrics, and baselines implemented in LagrangeBench (Toshev et al., 2023). We aim to show that CORGI boosts performance on a variety of CFD tasks without much overhead in training or inference time compared to a GNS.

In our results, we provide performance metrics as well as timing information. Our rollout times are given as the mean among all rollout steps during evaluation. Our training time is given as the total training time. Both times are reported in A100 seconds.

## 4.1 SETTINGS

**Datasets.** In all of our experiments, we build upon LagrangeBench (Toshev et al., 2023), a collection of seven Lagrangian benchmark datasets that cover a wide range of canonical incompressible flow configurations available under the MIT license. Each dataset is generated with a weakly–compressible Smoothed Particle Hydrodynamics (SPH) solver and, crucially for learning systems, every 100th solver step is stored, turning the raw simulator output into a temporal coarse-graining task (i.e. the model must advance the flow 100 physical time–steps at once). We elaborate on the main characteristics that are relevant for interpreting our quantitative results in App B.

Table 1: Metrics for each model on each dataset. Our results use the best checkpoint determined by $\text{MSE}_{20}$. The results are reported as $\mu \pm \sigma$, utilizing the entire test dataset. The distributions are generally positively skewed, and hence the errors are asymmetric. Due to outliers, the standard deviation may be larger than the mean.

| Dataset | Model | $\text{MSE}_{20}$ | $\text{MSE}_{E_{\text{kin}}}$ | Sinkhorn | Training time (s) | Inference time (s) |
|---|---|---|---|---|---|---|
| DAM-2D | GNS | $3.86 \times 10^{-5} \pm 2.86 \times 10^{-5}$ | $1.59 \times 10^{-4} \pm 2.38 \times 10^{-4}$ | $1.36 \times 10^{-5} \pm 1.71 \times 10^{-5}$ | $2.03 \times 10^{4}$ | $1.15 \times 10^{-2}$ |
| | SEGNN | $5.04 \times 10^{-5} \pm 3.81 \times 10^{-5}$ | $1.34 \times 10^{-4} \pm 1.98 \times 10^{-4}$ | $2.38 \times 10^{-5} \pm 2.93 \times 10^{-5}$ | $4.01 \times 10^{4}$ | $2.79 \times 10^{-2}$ |
| | CoRGI | $1.55 \times 10^{-5} \pm 1.51 \times 10^{-5}$ | $2.18 \times 10^{-5} \pm 3.55 \times 10^{-5}$ | $2.82 \times 10^{-6} \pm 2.32 \times 10^{-6}$ | $2.10 \times 10^{4}$ | $1.13 \times 10^{-2}$ |
| LDC-2D | GNS | $1.64 \times 10^{-5} \pm 2.32 \times 10^{-6}$ | $4.64 \times 10^{-7} \pm 2.90 \times 10^{-7}$ | $1.07 \times 10^{-6} \pm 2.76 \times 10^{-7}$ | $1.56 \times 10^{4}$ | $8.72 \times 10^{-3}$ |
| | SEGNN | $1.91 \times 10^{-5} \pm 2.42 \times 10^{-6}$ | $5.79 \times 10^{-7} \pm 4.20 \times 10^{-7}$ | $1.72 \times 10^{-6} \pm 4.06 \times 10^{-7}$ | $2.73 \times 10^{4}$ | $1.67 \times 10^{-2}$ |
| | CoRGI | $1.45 \times 10^{-5} \pm 2.24 \times 10^{-6}$ | $3.81 \times 10^{-7} \pm 2.43 \times 10^{-7}$ | $5.07 \times 10^{-7} \pm 7.46 \times 10^{-8}$ | $1.78 \times 10^{4}$ | $1.05 \times 10^{-2}$ |
| RPF-2D | GNS | $3.69 \times 10^{-6} \pm 7.31 \times 10^{-7}$ | $2.81 \times 10^{-5} \pm 3.23 \times 10^{-5}$ | $1.92 \times 10^{-7} \pm 6.90 \times 10^{-8}$ | $1.50 \times 10^{4}$ | $1.03 \times 10^{-2}$ |
| | SEGNN | $3.51 \times 10^{-6} \pm 7.63 \times 10^{-7}$ | $1.78 \times 10^{-5} \pm 1.93 \times 10^{-5}$ | $2.74 \times 10^{-7} \pm 1.05 \times 10^{-7}$ | $2.25 \times 10^{4}$ | $1.47 \times 10^{-2}$ |
| | CoRGI | $1.54 \times 10^{-6} \pm 4.03 \times 10^{-7}$ | $2.39 \times 10^{-6} \pm 2.45 \times 10^{-6}$ | $2.08 \times 10^{-8} \pm 4.50 \times 10^{-9}$ | $1.68 \times 10^{4}$ | $1.09 \times 10^{-2}$ |
| TGV-2D | GNS | $6.74 \times 10^{-6} \pm 9.75 \times 10^{-6}$ | $4.64 \times 10^{-7} \pm 1.16 \times 10^{-6}$ | $4.39 \times 10^{-7} \pm 5.57 \times 10^{-7}$ | $1.50 \times 10^{4}$ | $8.63 \times 10^{-3}$ |
| | SEGNN | $4.40 \times 10^{-6} \pm 6.89 \times 10^{-6}$ | $3.95 \times 10^{-7} \pm 1.09 \times 10^{-6}$ | $1.85 \times 10^{-7} \pm 2.43 \times 10^{-7}$ | $2.19 \times 10^{4}$ | $1.22 \times 10^{-2}$ |
| | CoRGI | $3.81 \times 10^{-6} \pm 5.62 \times 10^{-6}$ | $2.90 \times 10^{-7} \pm 7.68 \times 10^{-7}$ | $1.05 \times 10^{-7} \pm 8.74 \times 10^{-8}$ | $1.61 \times 10^{4}$ | $8.88 \times 10^{-3}$ |
| LDC-3D | GNS | $4.15 \times 10^{-5} \pm 2.98 \times 10^{-6}$ | $1.86 \times 10^{-8} \pm 1.53 \times 10^{-8}$ | $4.82 \times 10^{-7} \pm 1.91 \times 10^{-7}$ | $3.79 \times 10^{4}$ | $2.18 \times 10^{-2}$ |
| | SEGNN | $4.18 \times 10^{-5} \pm 3.00 \times 10^{-6}$ | $4.00 \times 10^{-8} \pm 2.67 \times 10^{-8}$ | $2.64 \times 10^{-7} \pm 9.79 \times 10^{-8}$ | $9.02 \times 10^{4}$ | $6.53 \times 10^{-2}$ |
| | CoRGI | $3.86 \times 10^{-5} \pm 2.84 \times 10^{-6}$ | $1.55 \times 10^{-8} \pm 9.02 \times 10^{-9}$ | $2.69 \times 10^{-7} \pm 9.87 \times 10^{-8}$ | $5.46 \times 10^{4}$ | $2.64 \times 10^{-2}$ |
| RPF-3D | GNS | $2.08 \times 10^{-5} \pm 1.73 \times 10^{-6}$ | $1.91 \times 10^{-6} \pm 1.89 \times 10^{-6}$ | $2.15 \times 10^{-7} \pm 4.69 \times 10^{-8}$ | $2.75 \times 10^{4}$ | $1.63 \times 10^{-2}$ |
| | SEGNN | $1.64 \times 10^{-5} \pm 1.47 \times 10^{-6}$ | $1.34 \times 10^{-6} \pm 1.51 \times 10^{-6}$ | $2.53 \times 10^{-7} \pm 8.97 \times 10^{-8}$ | $5.95 \times 10^{4}$ | $4.30 \times 10^{-2}$ |
| | CoRGI | $1.95 \times 10^{-5} \pm 1.80 \times 10^{-6}$ | $1.58 \times 10^{-6} \pm 1.51 \times 10^{-6}$ | $1.33 \times 10^{-7} \pm 2.24 \times 10^{-8}$ | $4.90 \times 10^{4}$ | $2.23 \times 10^{-2}$ |
| TGV-3D | GNS | $7.17 \times 10^{-3} \pm 7.00 \times 10^{-3}$ | $7.87 \times 10^{-2} \pm 8.05 \times 10^{-2}$ | $8.78 \times 10^{-5} \pm 8.33 \times 10^{-5}$ | $2.87 \times 10^{4}$ | $1.92 \times 10^{-2}$ |
| | SEGNN | $8.26 \times 10^{-3} \pm 1.77 \times 10^{-2}$ | $2.21 \times 10^{-3} \pm 4.42 \times 10^{-3}$ | $1.73 \times 10^{-4} \pm 2.49 \times 10^{-4}$ | $6.40 \times 10^{4}$ | $5.90 \times 10^{-2}$ |
| | CoRGI | $6.10 \times 10^{-3} \pm 5.96 \times 10^{-3}$ | $5.73 \times 10^{-2} \pm 5.43 \times 10^{-2}$ | $2.11 \times 10^{-5} \pm 1.60 \times 10^{-5}$ | $5.12 \times 10^{4}$ | $2.11 \times 10^{-2}$ |

**Computational Resources.** The experiments are done on Lambda's `gpu_8x_a100` cloud instances. The training time requirements for the above experiments are listed in the tables. In total, we estimate approximately 2500 A100 hours were used, including additional exploration not detailed here.

Our implementation details may be found in App C. They closely reflect those found in LagrangeBench, as in our experience they achieve reasonable results.

## 4.2 MAIN RESULTS

Using a GNS variant of CoRGI, on average we show a 57% improvement in accuracy compared to GNS, while only utilizing 13% more inference time and 31% more training time. Compared to SEGNN, CoRGI on average achieves a 49% improvement in accuracy, while also obtaining a 48% decrease in inference time and a 30% decrease in training time. The metrics are recorded in Table 1. We elaborate on SEGNN's performance in Sec 4.4, but we note here that SEGNN's inductive bias empirically seems to be helpful only on highly symmetrical and idealized scenarios, while also being significantly more expensive than GNS or CoRGI.

In our experiments, we compare against two established GNNs: GNS (Sanchez-Gonzalez et al., 2020) and SEGNN (Brandstetter et al., 2022). We choose these two models because they represent two different classes of GNN: non-equivariant (GNS) and equivariant (SEGNN). These are also identified as state of the art in the literature (Toshev et al., 2024), and to the authors' knowledge, remain so. We note that PaiNN (Schütt et al., 2021) and EGNN (Satorras et al., 2022), although both relatively efficient equivariant models (i.e. one without expensive Clebsch-Gordan tensor products), have been shown to be unsuitable for CFD (Toshev et al., 2023).

Qualitatively, we also see in Figure 2 that CoRGI overall maintains a much more uniform density than our baselines. Fluid incompressibility constraints (i.e. uniform density) are traditionally difficult to maintain in situations with high Reynolds numbers (i.e. turbulence), such as in dam break. We demonstrate that, even in highly turbulent situations, CoRGI provides strong performance in physical plausibility. As evident in Table 1, CoRGI performs an order of magnitude better compared to baselines in kinetic energy error and Sinkhorn divergence, both of which rely heavily on fluid incompressibility.

## 4.3 ADDITIONAL EXPERIMENTS

We focus on the RPF-2D dataset for our additional experiments, as it provides the clearest insight in where CoRGI's improvement originates. Detailed results are present in Appendix E. In summary, we

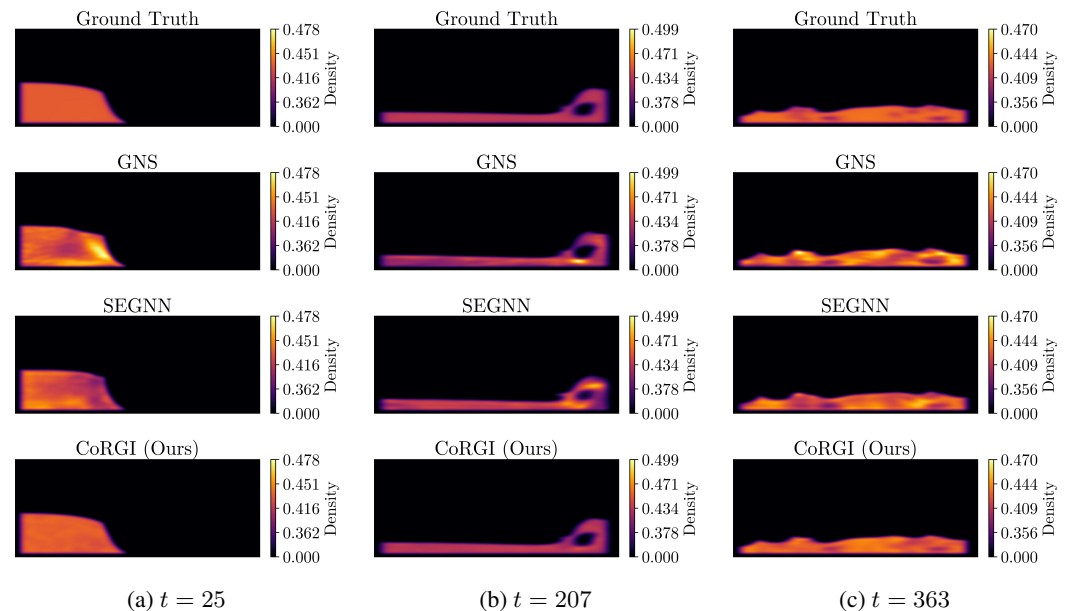

(a) $t = 25$          (b) $t = 207$          (c) $t = 363$

Figure 2: Kernel density estimation on DAM-2D ($t \in [\![0, 400]\!]$). Above, uniform coloring indicates adherence to fluid incompressibility. Qualitatively, CORGI maintains lower density variance under turbulent regimes.

find that the improvement offered by CORGI over GNS is robust to changes in number of message passing steps and graph connectivity. In other words, we expect that CORGI will improve upon GNS regardless of the specific budget for its graphical component.

Building upon this idea, in Fig 3, we illustrate the accuracy with regard to inference time for our metrics. Overall, CORGI demonstrates an average accuracy improvement of 47% given a time budget (chosen from the range in which both GNS and CORGI reside) over GNS. The metrics may be found in Table 2.

## 4.4 DISCUSSION

Overall, our experiments show that CORGI excels in its intended purpose: to capture global information. Table 1 shows that CORGI always performs the best in Sinkhorn divergence, except in LDC-3D where it is only 2% above SEGNN.

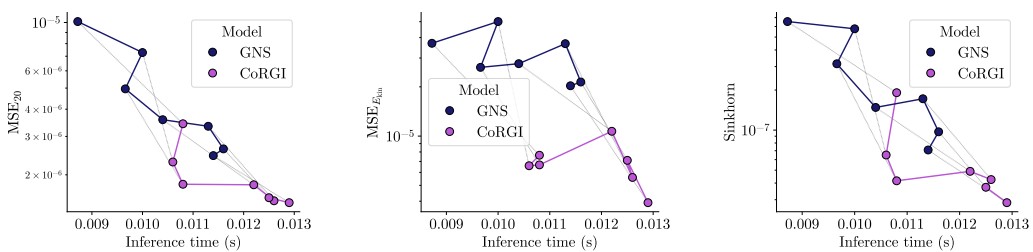

Figure 3: Plot of accuracy compared to inference time for GNS and CORGI on RPF-2D. Dotted lines pair points based on number of message passing steps. We adjust our allocated time budget in the above plots by adding or removing GNN message passing layers. Due to noise, the relationship is not empirically monotonic, but still demonstrates a trend of CORGI outperforming GNS for a given time budget.

Table 2: Performance based on number of message passing steps. We show that CORGI improves upon GNS regardless of the number of message passing steps. Moreover, CORGI often represents a better use of inference time budget, i.e. providing better accuracy for the same inference time.

| MP steps | Model | $\text{MSE}_{20}$ | $\text{MSE}_{E_{kin}}$ | Sinkhorn | Training time (s) | Inference time (s) |
|---|---|---|---|---|---|---|
| 4 | GNS | $1.01 \times 10^{-5} \pm 1.71 \times 10^{-6}$ | $3.69 \times 10^{-5} \pm 4.03 \times 10^{-5}$ | $6.54 \times 10^{-7} \pm 1.70 \times 10^{-7}$ | $1.14 \times 10^{4}$ | $8.72 \times 10^{-3}$ |
| | SEGNN | $1.17 \times 10^{-5} \pm 1.99 \times 10^{-6}$ | $3.55 \times 10^{-5} \pm 3.86 \times 10^{-5}$ | $7.89 \times 10^{-7} \pm 2.67 \times 10^{-7}$ | $1.36 \times 10^{4}$ | $8.99 \times 10^{-3}$ |
| | CORGI | $3.43 \times 10^{-6} \pm 6.22 \times 10^{-7}$ | $7.64 \times 10^{-6} \pm 8.50 \times 10^{-6}$ | $1.91 \times 10^{-7} \pm 5.17 \times 10^{-8}$ | $1.54 \times 10^{4}$ | $1.08 \times 10^{-2}$ |
| 6 | GNS | $7.28 \times 10^{-6} \pm 1.47 \times 10^{-6}$ | $5.01 \times 10^{-5} \pm 5.47 \times 10^{-5}$ | $5.77 \times 10^{-7} \pm 1.90 \times 10^{-7}$ | $1.35 \times 10^{4}$ | $1.00 \times 10^{-2}$ |
| | SEGNN | $7.02 \times 10^{-6} \pm 1.38 \times 10^{-6}$ | $2.68 \times 10^{-5} \pm 2.95 \times 10^{-5}$ | $5.18 \times 10^{-7} \pm 2.02 \times 10^{-7}$ | $1.73 \times 10^{4}$ | $1.14 \times 10^{-2}$ |
| | CORGI | $2.29 \times 10^{-6} \pm 5.19 \times 10^{-7}$ | $6.59 \times 10^{-6} \pm 7.46 \times 10^{-6}$ | $6.50 \times 10^{-8} \pm 2.07 \times 10^{-8}$ | $1.60 \times 10^{4}$ | $1.06 \times 10^{-2}$ |
| 8 | GNS | $4.96 \times 10^{-6} \pm 9.93 \times 10^{-7}$ | $2.63 \times 10^{-5} \pm 2.90 \times 10^{-5}$ | $3.15 \times 10^{-7} \pm 9.45 \times 10^{-8}$ | $1.40 \times 10^{4}$ | $9.66 \times 10^{-3}$ |
| | SEGNN | $4.83 \times 10^{-6} \pm 9.78 \times 10^{-7}$ | $1.97 \times 10^{-5} \pm 2.11 \times 10^{-5}$ | $3.80 \times 10^{-7} \pm 1.48 \times 10^{-7}$ | $2.07 \times 10^{4}$ | $1.35 \times 10^{-2}$ |
| | CORGI | $1.81 \times 10^{-6} \pm 4.40 \times 10^{-7}$ | $6.67 \times 10^{-6} \pm 7.47 \times 10^{-6}$ | $4.16 \times 10^{-8} \pm 1.39 \times 10^{-8}$ | $1.64 \times 10^{4}$ | $1.08 \times 10^{-2}$ |
| 10 | GNS | $3.58 \times 10^{-6} \pm 8.29 \times 10^{-7}$ | $2.77 \times 10^{-5} \pm 3.16 \times 10^{-5}$ | $1.48 \times 10^{-7} \pm 5.74 \times 10^{-8}$ | $1.56 \times 10^{4}$ | $1.04 \times 10^{-2}$ |
| | SEGNN | $3.40 \times 10^{-6} \pm 7.32 \times 10^{-7}$ | $1.78 \times 10^{-5} \pm 1.98 \times 10^{-5}$ | $3.23 \times 10^{-7} \pm 1.29 \times 10^{-7}$ | $2.29 \times 10^{4}$ | $1.50 \times 10^{-2}$ |
| | CORGI | $1.80 \times 10^{-6} \pm 3.77 \times 10^{-7}$ | $1.07 \times 10^{-5} \pm 1.34 \times 10^{-5}$ | $4.90 \times 10^{-8} \pm 1.37 \times 10^{-8}$ | $1.82 \times 10^{4}$ | $1.22 \times 10^{-2}$ |
| 12 | GNS | $3.34 \times 10^{-6} \pm 7.91 \times 10^{-7}$ | $3.67 \times 10^{-5} \pm 3.94 \times 10^{-5}$ | $1.72 \times 10^{-7} \pm 6.42 \times 10^{-8}$ | $1.66 \times 10^{4}$ | $1.13 \times 10^{-2}$ |
| | SEGNN | $2.47 \times 10^{-6} \pm 5.60 \times 10^{-7}$ | $1.88 \times 10^{-5} \pm 2.17 \times 10^{-5}$ | $1.88 \times 10^{-7} \pm 8.77 \times 10^{-8}$ | $2.59 \times 10^{4}$ | $1.70 \times 10^{-2}$ |
| | CORGI | $1.52 \times 10^{-6} \pm 4.44 \times 10^{-7}$ | $5.59 \times 10^{-6} \pm 6.48 \times 10^{-6}$ | $4.25 \times 10^{-8} \pm 2.07 \times 10^{-8}$ | $1.99 \times 10^{4}$ | $1.26 \times 10^{-2}$ |
| 14 | GNS | $2.63 \times 10^{-6} \pm 6.10 \times 10^{-7}$ | $2.14 \times 10^{-5} \pm 2.35 \times 10^{-5}$ | $9.71 \times 10^{-8} \pm 3.59 \times 10^{-8}$ | $1.78 \times 10^{4}$ | $1.16 \times 10^{-2}$ |
| | SEGNN | $2.05 \times 10^{-6} \pm 4.45 \times 10^{-7}$ | $1.13 \times 10^{-5} \pm 1.23 \times 10^{-5}$ | $1.61 \times 10^{-7} \pm 7.33 \times 10^{-8}$ | $2.96 \times 10^{4}$ | $1.92 \times 10^{-2}$ |
| | CORGI | $1.57 \times 10^{-6} \pm 3.18 \times 10^{-7}$ | $7.11 \times 10^{-6} \pm 7.82 \times 10^{-6}$ | $3.72 \times 10^{-8} \pm 1.18 \times 10^{-8}$ | $2.01 \times 10^{4}$ | $1.25 \times 10^{-2}$ |
| 16 | GNS | $2.45 \times 10^{-6} \pm 5.48 \times 10^{-7}$ | $2.03 \times 10^{-5} \pm 2.20 \times 10^{-5}$ | $7.09 \times 10^{-8} \pm 2.67 \times 10^{-8}$ | $1.84 \times 10^{4}$ | $1.14 \times 10^{-2}$ |
| | SEGNN | $1.66 \times 10^{-6} \pm 3.46 \times 10^{-7}$ | $9.40 \times 10^{-6} \pm 1.05 \times 10^{-5}$ | $1.29 \times 10^{-7} \pm 6.03 \times 10^{-8}$ | $3.37 \times 10^{4}$ | $2.16 \times 10^{-2}$ |
| | CORGI | $1.49 \times 10^{-6} \pm 2.85 \times 10^{-7}$ | $3.92 \times 10^{-6} \pm 4.72 \times 10^{-6}$ | $2.85 \times 10^{-8} \pm 9.06 \times 10^{-9}$ | $2.05 \times 10^{4}$ | $1.29 \times 10^{-2}$ |

Intuitively, aggregating information on a global scale allows a model to better learn global dynamics. While $\text{MSE}_{20}$ and $\text{MSE}_{E_{kin}}$ are primarily local errors (i.e. $\text{MSE}_{20}$ measures pointwise positional error and $\text{MSE}_{E_{kin}}$ can be interpreted as pointwise energy error), Sinkhorn divergence measures the distance between particle distributions themselves, and hence is global in nature. Thus, it is not necessarily expected for CORGI to outperform every model in every metric.

Nonetheless, the experiments in which SEGNN showed better performance are somewhat irreflective of real-world performance. For instance, RPF-3D is far more symmetrical than the typical real-world prediction task, and also has a low Reynolds number, and hence, an equivariant model may naturally learn such dynamics better. However, in more complex scenarios such as LDC-3D, CORGI retains superior accuracy. Given our performance on these types of datasets, we believe CORGI to be better for real-world use, including irregular or turbulent flows, at a fraction of the cost of SEGNN.

We additionally demonstrate the ability of CORGI to better utilize training and inference time budget, i.e. that it is more useful to focus on both local and global interactions rather than exclusively to deepen the GNN. However, in some cases, it is still useful to integrate an equivariant inductive bias. We intend to explore this incorporation in future work, and we discuss this and other limitations in Appendix F.

## 5 CONCLUSION

In this paper, we introduce Convolutional Residual Global Interactions (CORGI), a lightweight augmentation on GNN-based simulators which incorporates global knowledge into GNN predictions for hydrodynamics. By expanding the receptive field to a global scale, we overcome the subsonic propagation speed of a traditional GNN, and enable directly modeling nonlocal effects such as advection. We reduce GNS's error by 57% out of the box, and when restricted to the same inference time budget, still demonstrate a 47% improvement in accuracy. CORGI through our testing appears as a highly promising and efficient architecture for computational fluid dynamics compared to existing arts in neural emulation.

## REPRODUCIBILITY STATEMENT

All required settings to reproduce experiments are detailed in the paper. The dataset used is described in App B, and our hyperparameters are detailed in App C. All other settings are included as part of the relevant figures or tables. Code is provided as supplementary material.

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

## A  THEORETICAL ANALYSIS

In this section, we provide some further discussions to support the necessity of CORGI, and prove Theorem 1. The basis of our analysis is the Courant-Friedrich-Lewy condition (Courant et al., 1967):

$$\frac{v\Delta_t}{\Delta_x} \le 1, \tag{5}$$

where $v \in \mathbb{R}$ is the speed of sound, $\Delta_x \in \mathbb{R}$ is the length traveled per step (i.e. receptive field), and $\Delta_t \in \mathbb{R}$ is the length of a time step. We use this as a benchmark for the required speed of information propagation within the model.

### A.1 GNN MESSAGE PASSING IS TOO SLOW

Interpreted in our setting, we can impose an upper bound on $v\Delta_t$ as $||b_{\max} - b_{\min}||_2$ where $b_{\max}$ and $b_{\min}$ are the maximal and minimal coordinates. We impose such an upper bound since this is the furthest distance data may travel in between time frames. In our analyses, we set the speed of sound to exactly this upper bound, as we interpret the speed of sound as some arbitrarily large value. It is evident that GNNs such as GNS or SEGNN will violate this condition with few message passing steps in this worst-case setting. Given a connectivity radius $r \in \mathbb{R}$ and $L \in \mathbb{N}$ message passing steps, we have from the analyses below that $||b_{\max} - b_{\min}||_2/(rL)$ must be less than 1 for effective information exchange. However, in typical settings, e.g. in LagrangeBench, the baseline GNNs yield Courant numbers much larger than 1:

Table 3: LagrangeBench Courant numbers (with $L = 10$)

| Dataset | Dimensions | Connectivity radius | Courant number |
|---------|-----------|--------------------|----------------| 
| DAM-2D | $5.586 \times 2.22$ | 0.029 | 20.73 |
| LDC-2D | $1.12 \times 1.12$ | 0.029 | 5.46 |
| RPF-2D | $1.0 \times 2.0$ | 0.036 | 6.21 |
| TGV-2D | $1.0 \times 1.0$ | 0.029 | 4.88 |
| LDC-3D | $1.25 \times 1.25 \times 0.5$ | 0.06 | 3.06 |
| RPF-3D | $1.0 \times 2.0 \times 0.5$ | 0.072 | 3.18 |
| TGV-3D | $2\pi \times 2\pi \times 2\pi$ | 0.46 | 2.37 |

### A.2 PROOF OF THEOREM 1: CORGI ENABLES FAST INFORMATION PROPAGATION

*Proof.* First, we consider the furthest distance information can travel with $L$ layers of GNN message passing (MP). For a starting node $v \in \mathcal{V}$, we denote nodes that are of degree at most $d$ (i.e., $d$-hop) from $v$ as $\mathcal{N}_d(v)$, where note that $\mathcal{N}_1(v) = \{v'|(v,v') \in \mathcal{E}\}$ are simply the neighbors of $v$. Then with $L$ MP layers, the information in $v$ can only propagate to nodes within $\mathcal{N}_L(v)$. Since the graph is constructed with radius $r$, the maximum distance of this propagation is $rL$.

Next, from (5) we have the length traveled per step is $\Delta_x \geq v\Delta_t$, and so when $v\Delta_t > rL$, there is a sizable gap between the information propagation distance of the GNN and the minimum required distance per step, which CORGI seeks to address. For our model, assume the grid constructed for CNNs at the highest resolution level has unit length $d_{\min}$ for each cell, while the lowest resolution level has unit length $d_{\max}$, then from the nature of CNN pooling, the number of layers must be at least $\Omega\big(\log(d_{\max}/d_{\min})\big)$ (in the case of $2 \times 2$ pooling in our case, the logarithm takes base 2).

Now we analyze the required magnitudes for $d_{\min}$ and $d_{\max}$, respectively. For $d_{\min}$, since the grid scatter operation pools information over all nodes within a unit cell, the feature value for CNN represents the *aggregated, averaged* information within the region of the unit cell. Therefore for each node within the cell to properly *send* and *receive* their more fine-grained individual information, respectively *before* and *after* the CORGI module, we would require $d_{\min} = \mathcal{O}(rL)$, or the unit cell length to not exceed the information propagation distance by the GNN in order of magnitude. For $d_{\max}$, we see the final layer of CNN allows interaction over maximum distance $C \times d_{\max}$, where $C$ is a constant associated with the CNN hyperparameters, such as kernel size and stride. Therefore we must have $d_{\max} = \Omega(v\Delta_t)$ to allow for the modeling of interaction on par with the Courant-Friedrich-Lewy condition (5).

Combining everything, we finally have the number of CNN layers within CORGI should be at least

$$\Omega\big(\log(d_{\max}/d_{\min})\big) = \Omega\big(\log(v\Delta_t/rL)\big).$$

$\square$

## B DATASETS

In all of our experiments, we build upon LagrangeBench (Toshev et al., 2023), a collection of seven Lagrangian benchmark datasets that cover a wide range of canonical incompressible flow configurations available under the MIT license. Each dataset is generated with a weakly–compressible

Smoothed Particle Hydrodynamics (SPH) solver and, crucially for learning systems, every 100th solver step is stored, turning the raw simulator output into a temporal coarse-graining task (i.e. the model must advance the flow 100 physical time–steps at once). Below we summarize the main characteristics that are relevant for interpreting our quantitative results.

Table 4: Physical characteristics

| Geometry | Re | Boundary conditions | Key challenge |
|---|---|---|---|
| Taylor-Green vortex | 100 (2D) 50 (3D) | fully periodic | rapid kinetic energy decay, symmetry preservation |
| Reverse Poiseuille flow | 10 | periodic, body-force driven | stationary shear layers with spatially varying forcing |
| Lid-driven cavity | 100 | moving lid, no-slip walls | accurate wall treatment, vortex formation |
| Dam break | 40000 | free surface, walls | highly nonlinear free surface evolution |

The Reynolds numbers in the table correspond to the effective Re in the SPH solver settings. 3D variants use the same physics as their 2D counterparts, simply extruded in the out-of-plane direction (with periodicity enforced in $z$).

Table 5: Numerical resolution and splits

| Datasets | # particles | Traj. length (train / val / test) | $\Delta t$ | Convolution res. |
|---|---|---|---|---|
| TGV-2D | 2500 | 126 / 63 / 63 | 0.04 | $32 \times 32$ |
| RPF-2D | 3200 | 20000 / 10000 / 10000 | 0.04 | $32 \times 64$ |
| LDC-2D | 2708 | 10000 / 5000 / 5000 | 0.04 | $32 \times 32$ |
| DAM-2D | 5740 | 401 / 200 / 200 | 0.03 | $80 \times 32$ |
| TGV-3D | 8000 | 61 / 31 / 31 | 0.5 | $32 \times 32 \times 32$ |
| RPF-3D | 8000 | 10000 / 5000 / 5000 | 0.1 | $32 \times 64 \times 16$ |
| LDC-3D | 8160 | 10000 / 5000 / 5000 | 0.09 | $40 \times 40 \times 16$ |

For statistically stationary cases (RPF, LDC) the simulator is run until equilibrium and a single long trajectory is partitioned into the above splits; for transient cases (TGV, DAM) multiple independent realizations are generated.

The step size $\Delta t$ refers to the physical time between two stored frames (i.e. one ML training step equals 100 SPH sub-steps and corresponds to $\Delta t$).

The collection was chosen to expose learning algorithms to a broad span of flow phenomena: decay to rest, driven steady states, wall-bounded recirculation, and violent free-surface motion; and progressively harder geometric settings (from fully periodic boxes to complex wall topologies). Moreover, by subsampling at $100\times$ the solver step, the benchmark explicitly tests a model's ability to extrapolate in time, which is a key requirement for practical surrogate solvers in engineering applications.

## C  IMPLEMENTATION DETAILS

**Pipeline.**  In training and evaluation, we first isolate a subtrajectory of a suitable length in every trajectory we use. The length of this subtrajectory is determined by how long our historical inputs are and how many rollout steps we take. In our case, we use a history length of 6, and we do 20 rollout steps, as these are the defaults provided by LagrangeBench, and hence we aim for a trajectory length of 26. When initially predicting a trajectory, we start with 6 ground truth states, which are aggregated into suitable features (namely position and velocity, detailed in Sec 2.3) and given to the model. At each time step, we recompute the neighbor graph. Then, we drop the least recent frame and append the next frame generated by the model's prediction via symplectic Euler integration.

**Learning Parameters.** All models were assessed with exponential decay. The initial learning rate is $5 \times 10^{-4}$ and the final learning rate is $1 \times 10^{-6}$. We use a decay rate of 0.1 and $1 \times 10^5$ decay steps. These are the same as in LagrangeBench. For GNS with and without CORGI, we use a batch size of 4. In our experience, SEGNN had better performance with a batch size of 1 using the same learning rates, hence we report its results with the batch size of 1. All models used 10 message passing layers unless stated otherwise.

**Hyperparameters.** For our convolutional layers, we use a feature plan of $(128, 256, 512)$, as it has a relatively good performance with low overhead compared to GNS. We choose our resolution to ensure the receptive field of each cell is roughly global. We attempt to justify these choices in Appendix E. The specific resolutions are given in Appendix B. In our main results, all of the models are assessed with 10 message passing layers, and graphical components all have a hidden dimension of 128; these are the defaults from LagrangeBench, and in our experience, offer reasonable performance.

**Metrics.** To assess how well a surrogate advances a particle system over many solver steps, we adopt three error measures: mean-squared position error, sinkhorn distance and kinetic energy MSE (See Appendix D for full description). All metrics are computed on rollouts of 20 steps and are averaged over every stored frame, particle and test trajectory, with the checkpoint chosen by the lowest validation $\text{MSE}_{20}$. When training, our loss is the MSE between the predicted positions and the ground truth predictions detailed in Appendix D.1.

# D  EVALUATION METRICS

To assess how well a surrogate advances a particle system over many solver steps, we adopt exactly the three error measures proposed by LagrangeBench. All metrics are computed on full roll-outs of length $n$ steps (we report results for $n = 20$) and are averaged over every stored frame, particle and test trajectory; the quoted numbers are finally averaged over three independent training runs, with the checkpoint chosen by the lowest validation $\text{MSE}_{20}$.

## D.1  MEAN-SQUARED POSITION ERROR

We track the $\mathcal{L}_2$ error between predicted and ground truth particle positions, aggregated over an $n$-step rollout:

$$\text{MSE}_n = \frac{1}{Nn} \sum_{k=1}^{n} \sum_{i=1}^{N} \|x_i^{(t+k)} - \hat{x}_i^{(t+k)}\|_2^2$$

## D.2  SINKHORN DISTANCE

Pure position errors ignore that particles are interchangeable. We thus use the entropy-regularized optimal transport distance (i.e. Sinkhorn loss (Cuturi, 2013))

$$\mathcal{S}_\varepsilon(\mathbf{x}, \hat{\mathbf{x}}) = \min_{\Gamma \in \Pi(\mathbf{u}, \mathbf{v})} \langle \Gamma, C \rangle + \varepsilon \text{KL}(\Gamma \| \mathbf{u}\mathbf{v}^\intercal)$$

where $C_{ij} = \|\mathbf{x}_i - \hat{\mathbf{x}}_i\|_2^2$ is a cost matrix and $\Pi(\mathbf{u}, \mathbf{v})$ is the set of couplings whose row and column sums match the source mass marginal distribution $\mathbf{u}$ and the target mass marginal distribution $\mathbf{v}$. In our case, each row and column in the coupling matrix must sum to $\frac{1}{N}$, i.e. the mass of a particle.

## D.3  KINETIC ENERGY MSE

To capture global physical consistency we compare the system's kinetic energy time series

$$T(t) = \frac{1}{2} \sum_{i=1}^{N} m_i \|\dot{x}_i^{(t)}\|^2$$

and report the mean-squared error between predicted and reference curves over the rollout window.

## D.4 KERNEL-BASED METRICS

In order to assess the physical plausibility of CORGI's improvements, we also employ some kernel-based metrics using the quintic spline (Morris et al., 1997).

Let the smoothing length $h$ be the average distance between particles. Then, the quintic spline is defined as

$$W(r) = C_d h^{-d} (\max(0, 3 - \tfrac{r}{h})^5 - 6 \max(0, 2 - \tfrac{r}{h})^3 + 15 \max(0, 1 - \tfrac{r}{h}))$$

where $C_2 = \frac{7}{478\pi}$ and $C_3 = \frac{1}{120\pi}$. Using $\mathbf{r}_{ij} \in \mathbb{R}^d$ to represent the displacement between two particles $i$ and $j$, we write the directional gradient

$$\nabla W_{ij} = \frac{\mathbf{r}_{ij}}{||\mathbf{r}_{ij}||} \frac{d}{d||\mathbf{r}_{ij}||} W(||\mathbf{r}_{ij}||)$$

For divergence, we define the divergence at time step $t$ for particle $i$ to be

$$\frac{\sum_j (\dot{\mathbf{x}}_j^t - \dot{\mathbf{x}}_i^t) \cdot \nabla W_{ij}}{\sum_j W(||\mathbf{x}_j - \mathbf{x}_i||)}$$

For vorticity, we perform a similar computation to divergence for the 3 dimensional case:

$$\frac{\sum_j (\dot{\mathbf{x}}_j^t - \dot{\mathbf{x}}_i^t) \times \nabla W_{ij}}{\sum_j W(||\mathbf{x}_j - \mathbf{x}_i||)}$$

For the 2 dimensional case, embed the vector space in 3 dimensions, setting the third coordinate to 0.

For both divergence and vorticity, the reported metrics are the MSE over all particles and time steps, where error is defined to be the $\mathcal{L}_2$ distance.

Table 6: Kernel based metrics.

| Dataset | Model | Divergence error | Vorticity error |
|---------|-------|------------------|-----------------|
| DAM-2D | GNS | $1.20 \pm 1.01$ | $2.73 \pm 2.76$ |
| | CORGI | $1.04 \pm 9.93 \times 10^{-1}$ | $2.44 \pm 2.80$ |
| LDC-2D | GNS | $7.76 \times 10^{-1} \pm 4.76 \times 10^{-2}$ | $1.64 \pm 1.31 \times 10^{-1}$ |
| | CORGI | $6.54 \times 10^{-1} \pm 4.19 \times 10^{-2}$ | $1.55 \pm 1.23 \times 10^{-1}$ |
| RPF-2D | GNS | $4.11 \times 10^{-2} \pm 2.69 \times 10^{-3}$ | $9.51 \times 10^{-2} \pm 9.80 \times 10^{-3}$ |
| | CORGI | $2.97 \times 10^{-2} \pm 3.19 \times 10^{-3}$ | $7.41 \times 10^{-2} \pm 1.10 \times 10^{-2}$ |
| TGV-2D | GNS | $7.93 \times 10^{-2} \pm 1.23 \times 10^{-1}$ | $2.17 \times 10^{-1} \pm 3.39 \times 10^{-1}$ |
| | CORGI | $8.38 \times 10^{-2} \pm 1.27 \times 10^{-1}$ | $2.23 \times 10^{-1} \pm 3.44 \times 10^{-1}$ |
| LDC-3D | GNS | $1.16 \times 10^{-1} \pm 3.63 \times 10^{-3}$ | $3.99 \times 10^{-1} \pm 1.44 \times 10^{-2}$ |
| | CORGI | $1.12 \times 10^{-1} \pm 4.03 \times 10^{-3}$ | $3.83 \times 10^{-1} \pm 1.54 \times 10^{-2}$ |
| RPF-3D | GNS | $4.50 \times 10^{-2} \pm 1.35 \times 10^{-3}$ | $1.16 \times 10^{-1} \pm 3.66 \times 10^{-3}$ |
| | CORGI | $4.39 \times 10^{-2} \pm 1.47 \times 10^{-3}$ | $1.13 \times 10^{-1} \pm 3.96 \times 10^{-3}$ |
| TGV-3D | GNS | $4.40 \times 10^{-3} \pm 4.35 \times 10^{-3}$ | $1.66 \times 10^{-2} \pm 1.63 \times 10^{-2}$ |
| | CORGI | $2.82 \times 10^{-3} \pm 2.78 \times 10^{-3}$ | $1.15 \times 10^{-2} \pm 1.13 \times 10^{-2}$ |

Based on our divergence errors, CORGI typically respects fluid incompressibility better than GNS. Moreover, based on the vorticity errors, our results in Table 1 were achieved without arbitrary dampening of angular momentum. Hence, CORGI retains physical realism comparable or superior to GNS while offering superior pointwise accuracy.

## E ADDITIONAL EXPERIMENTS

### E.1 ABLATION

We wish to show that our intuitions regarding the architecture of CORGI are empirically justified. Hence, we conduct several other experiments in order to gain intuition on the performance character-

istics of CORGI. These experiments are mostly done with RPF-2D, with another dataset used only if otherwise stated.

### E.1.1 SKIP CONNECTIONS

We assess which skip connections are most responsible for CORGI's performance based on convolution depth. We denote the presence of a skip with either a $\top$ (there is a skip connection) or a $\bot$ (there is not a skip connection).

Table 7: Performance based on skip connections

| Skips | $\text{MSE}_{20}$ | $\text{MSE}_{E_{\text{kin}}}$ | Sinkhorn | Training time | Inference time |
|---|---|---|---|---|---|
| $\bot, \bot, \bot$ | $1.76 \times 10^{-6} \pm 4.44 \times 10^{-7}$ | $1.18 \times 10^{-5} \pm 1.34 \times 10^{-5}$ | $8.23 \times 10^{-8} \pm 3.64 \times 10^{-8}$ | $1.84 \times 10^4$ | $1.21 \times 10^{-2}$ |
| $\bot, \bot, \top$ | $1.60 \times 10^{-6} \pm 4.07 \times 10^{-7}$ | $4.67 \times 10^{-6} \pm 4.56 \times 10^{-6}$ | $3.34 \times 10^{-8} \pm 1.26 \times 10^{-8}$ | $1.71 \times 10^4$ | $1.10 \times 10^{-2}$ |
| $\bot, \top, \bot$ | $1.68 \times 10^{-6} \pm 4.41 \times 10^{-7}$ | $8.54 \times 10^{-6} \pm 1.03 \times 10^{-5}$ | $4.41 \times 10^{-8} \pm 1.83 \times 10^{-8}$ | $1.78 \times 10^4$ | $1.13 \times 10^{-2}$ |
| $\bot, \top, \top$ | $1.55 \times 10^{-6} \pm 4.28 \times 10^{-7}$ | $6.21 \times 10^{-6} \pm 7.10 \times 10^{-6}$ | $3.11 \times 10^{-8} \pm 1.20 \times 10^{-8}$ | $1.77 \times 10^4$ | $1.13 \times 10^{-2}$ |
| $\top, \bot, \bot$ | $2.08 \times 10^{-6} \pm 4.56 \times 10^{-7}$ | $1.78 \times 10^{-5} \pm 1.95 \times 10^{-5}$ | $8.91 \times 10^{-8} \pm 3.94 \times 10^{-8}$ | $1.83 \times 10^4$ | $1.21 \times 10^{-2}$ |
| $\top, \bot, \top$ | $1.69 \times 10^{-6} \pm 4.41 \times 10^{-7}$ | $5.51 \times 10^{-6} \pm 5.70 \times 10^{-6}$ | $4.05 \times 10^{-8} \pm 1.48 \times 10^{-8}$ | $1.73 \times 10^4$ | $1.11 \times 10^{-2}$ |
| $\top, \top, \bot$ | $2.27 \times 10^{-6} \pm 4.14 \times 10^{-7}$ | $1.63 \times 10^{-5} \pm 1.76 \times 10^{-5}$ | $7.53 \times 10^{-8} \pm 2.50 \times 10^{-8}$ | $1.86 \times 10^4$ | $1.24 \times 10^{-2}$ |
| $\top, \top, \top$ | $1.58 \times 10^{-6} \pm 4.10 \times 10^{-7}$ | $4.55 \times 10^{-6} \pm 4.71 \times 10^{-6}$ | $4.41 \times 10^{-8} \pm 1.47 \times 10^{-8}$ | $1.92 \times 10^4$ | $1.25 \times 10^{-2}$ |

In Table 7, we assess the importance of skip connections at each depth. From left to right, our skip notation denotes the presence of skips from the finest to coarsest resolutions in a CORGI with a global convolution module of depth 3. In this table, it represents both the A and B skip connections of a particular depth, as denoted in Fig 1.

The data show that the skip connections at the coarsest resolution provide the most significant improvement on performance, followed by the middle layer. Curiously, having a skip connection for the finest layer actually hurts performance. We hypothesize this degradation to be caused by the additional learning inertia: in the finest resolution, the skip connection concatenates information both before and after only one double convolution.

We also note that the training and inference times are not significantly affected by the presence of these skip connections.

Table 8: Performance based on skip connections

| Skips | $\text{MSE}_{20}$ | $\text{MSE}_{E_{\text{kin}}}$ | Sinkhorn | Training time | Inference time |
|---|---|---|---|---|---|
| $\bot, \bot$ | $1.76 \times 10^{-6} \pm 4.44 \times 10^{-7}$ | $1.18 \times 10^{-5} \pm 1.34 \times 10^{-5}$ | $8.23 \times 10^{-8} \pm 3.64 \times 10^{-8}$ | $1.84 \times 10^4$ | $1.21 \times 10^{-2}$ |
| $\bot, \top$ | $1.77 \times 10^{-6} \pm 3.58 \times 10^{-7}$ | $1.11 \times 10^{-5} \pm 1.36 \times 10^{-5}$ | $6.99 \times 10^{-8} \pm 2.21 \times 10^{-8}$ | $1.56 \times 10^4$ | $9.35 \times 10^{-3}$ |
| $\top, \bot$ | $1.86 \times 10^{-6} \pm 3.73 \times 10^{-7}$ | $9.43 \times 10^{-6} \pm 1.07 \times 10^{-5}$ | $5.97 \times 10^{-8} \pm 2.08 \times 10^{-8}$ | $1.61 \times 10^4$ | $9.42 \times 10^{-3}$ |
| $\top, \top$ | $1.58 \times 10^{-6} \pm 4.10 \times 10^{-7}$ | $4.55 \times 10^{-6} \pm 4.71 \times 10^{-6}$ | $4.41 \times 10^{-8} \pm 1.47 \times 10^{-8}$ | $1.92 \times 10^4$ | $1.25 \times 10^{-2}$ |

We additionally show the necessity of both sets of skip connections (i.e. both A and B in Fig 1) in Table 8. Although neither one provides much of a boost in accuracy on their own, we demonstrate that including both sets of skip connections is the most optimal, without a significant overhead in training or inference time.

### E.1.2 CONVOLUTION

In Table 13, we show the importance of our global convolution module, in particular comparing versions with and without any convolution operations. For a depth of 0, we directly use a GNS as a baseline. Even after adding a single convolutional level, we see drastic improvements across all metrics. Adding on more depth generally improves performance, however we see there are diminishing returns past 3 resolution levels, as the receptive field is nearly global at that point, and hence we fulfill the conditions from App A.

Moreover, we also assess a convolution-only architecture (Rochman-Sharabi et al., 2025) below in Table 9. We omit a comparison on the RPF-3D dataset due to technical difficulties. However, across all other datasets, CORGI performs significantly better.

Table 9: Performance using only a U-Net, i.e. NeuralMPM. As LagrangeBench computes the neighbor graph regardless of the model's dependency on it, and we conduct this experiment as an ablation rather as a standalone experiment, our reported inference and training times include the time to construct the graph.

| Dataset | $MSE_{20}$ | $MSE_{E_{kin}}$ | Sinkhorn | Training time (s) | Inference time (s) |
|---|---|---|---|---|---|
| DAM-2D | $4.95 \times 10^{-5} \pm 5.91 \times 10^{-5}$ | $6.48 \times 10^{-5} \pm 2.05 \times 10^{-4}$ | $1.27 \times 10^{-5} \pm 2.01 \times 10^{-5}$ | $1.41 \times 10^{4}$ | $2.01 \times 10^{-2}$ |
| LDC-2D | $2.57 \times 10^{-5} \pm 2.05 \times 10^{-6}$ | $2.29 \times 10^{-6} \pm 1.49 \times 10^{-6}$ | $3.13 \times 10^{-7} \pm 5.19 \times 10^{-8}$ | $1.27 \times 10^{4}$ | $1.40 \times 10^{-2}$ |
| RPF-2D | $4.87 \times 10^{-5} \pm 3.58 \times 10^{-6}$ | $4.72 \times 10^{-5} \pm 5.08 \times 10^{-5}$ | $1.08 \times 10^{-7} \pm 3.19 \times 10^{-8}$ | $1.40 \times 10^{4}$ | $2.31 \times 10^{-2}$ |
| TGv-2D | $1.87 \times 10^{-5} \pm 2.57 \times 10^{-5}$ | $1.46 \times 10^{-6} \pm 3.11 \times 10^{-6}$ | $-6.63 \times 10^{-9} \pm 6.10 \times 10^{-7}$ | $1.40 \times 10^{4}$ | $2.08 \times 10^{-2}$ |
| LDC-3D | $1.03 \times 10^{-4} \pm 3.84 \times 10^{-6}$ | $1.16 \times 10^{-7} \pm 5.04 \times 10^{-8}$ | $5.23 \times 10^{-6} \pm 6.54 \times 10^{-7}$ | $2.03 \times 10^{4}$ | $1.68 \times 10^{-2}$ |
| TGV-3D | $1.61 \times 10^{-2} \pm 1.49 \times 10^{-2}$ | $9.43 \times 10^{-1} \pm 9.52 \times 10^{-1}$ | $3.28 \times 10^{-4} \pm 2.71 \times 10^{-4}$ | $1.48 \times 10^{4}$ | $2.07 \times 10^{-2}$ |

### E.1.3 HISTORY

We additionally assess the impact of history length in model performance. Our motivation is that macroscopic physics is approximately deterministic, i.e. the future state of a system should be entirely determinable from a state and a set of physical laws. Nonetheless, in approximations, it may be helpful to include historical information in order to numerically approximate acceleration, jerk, etc.

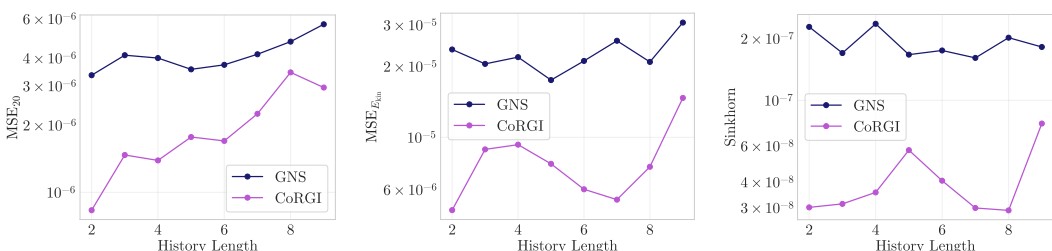

Figure 4: Plot of accuracy compared to history for GNS and CORGI on RPF-2D.

Table 10: Performance based on history length

| History | Model | $MSE_{20}$ | $MSE_{E_{kin}}$ | Sinkhorn | Training time | Inference time |
|---|---|---|---|---|---|---|
| 2 | GNS | $3.35 \times 10^{-6} \pm 8.33 \times 10^{-7}$ | $2.36 \times 10^{-5} \pm 2.50 \times 10^{-5}$ | $2.28 \times 10^{-7} \pm 7.71 \times 10^{-8}$ | $1.56 \times 10^{4}$ | $1.07 \times 10^{-2}$ |
| | CORGI | $8.31 \times 10^{-7} \pm 2.22 \times 10^{-7}$ | $4.91 \times 10^{-6} \pm 5.64 \times 10^{-6}$ | $3.01 \times 10^{-8} \pm 9.68 \times 10^{-9}$ | $1.88 \times 10^{4}$ | $1.22 \times 10^{-2}$ |
| 3 | GNS | $4.12 \times 10^{-6} \pm 9.69 \times 10^{-7}$ | $2.05 \times 10^{-5} \pm 2.42 \times 10^{-5}$ | $1.70 \times 10^{-7} \pm 5.59 \times 10^{-8}$ | $1.49 \times 10^{4}$ | $9.92 \times 10^{-3}$ |
| | CORGI | $1.47 \times 10^{-6} \pm 3.10 \times 10^{-7}$ | $8.89 \times 10^{-6} \pm 9.98 \times 10^{-6}$ | $3.13 \times 10^{-8} \pm 8.70 \times 10^{-9}$ | $1.83 \times 10^{4}$ | $1.19 \times 10^{-2}$ |
| 4 | GNS | $4.00 \times 10^{-6} \pm 8.87 \times 10^{-7}$ | $2.19 \times 10^{-5} \pm 2.48 \times 10^{-5}$ | $2.36 \times 10^{-7} \pm 7.91 \times 10^{-8}$ | $1.53 \times 10^{4}$ | $1.01 \times 10^{-2}$ |
| | CORGI | $1.39 \times 10^{-6} \pm 2.97 \times 10^{-7}$ | $9.31 \times 10^{-6} \pm 1.15 \times 10^{-5}$ | $3.56 \times 10^{-8} \pm 1.12 \times 10^{-8}$ | $1.71 \times 10^{4}$ | $1.09 \times 10^{-2}$ |
| 5 | GNS | $3.56 \times 10^{-6} \pm 8.68 \times 10^{-7}$ | $1.75 \times 10^{-5} \pm 2.03 \times 10^{-5}$ | $1.67 \times 10^{-7} \pm 5.98 \times 10^{-8}$ | $1.56 \times 10^{4}$ | $1.08 \times 10^{-2}$ |
| | CORGI | $1.77 \times 10^{-6} \pm 3.52 \times 10^{-7}$ | $7.72 \times 10^{-6} \pm 8.78 \times 10^{-6}$ | $5.72 \times 10^{-8} \pm 1.71 \times 10^{-8}$ | $1.75 \times 10^{4}$ | $1.12 \times 10^{-2}$ |
| 6 | GNS | $3.73 \times 10^{-6} \pm 7.99 \times 10^{-7}$ | $2.11 \times 10^{-5} \pm 2.32 \times 10^{-5}$ | $1.75 \times 10^{-7} \pm 6.39 \times 10^{-8}$ | $1.59 \times 10^{4}$ | $1.08 \times 10^{-2}$ |
| | CORGI | $1.70 \times 10^{-6} \pm 3.55 \times 10^{-7}$ | $6.02 \times 10^{-6} \pm 7.26 \times 10^{-6}$ | $4.06 \times 10^{-8} \pm 1.12 \times 10^{-8}$ | $1.73 \times 10^{4}$ | $1.10 \times 10^{-2}$ |
| 7 | GNS | $4.16 \times 10^{-6} \pm 9.99 \times 10^{-7}$ | $2.57 \times 10^{-5} \pm 2.76 \times 10^{-5}$ | $1.61 \times 10^{-7} \pm 5.48 \times 10^{-8}$ | $1.50 \times 10^{4}$ | $1.00 \times 10^{-2}$ |
| | CORGI | $2.25 \times 10^{-6} \pm 5.67 \times 10^{-7}$ | $5.44 \times 10^{-6} \pm 5.91 \times 10^{-6}$ | $2.99 \times 10^{-8} \pm 1.11 \times 10^{-8}$ | $1.81 \times 10^{4}$ | $1.15 \times 10^{-2}$ |
| 8 | GNS | $4.74 \times 10^{-6} \pm 8.94 \times 10^{-7}$ | $2.09 \times 10^{-5} \pm 2.30 \times 10^{-5}$ | $2.02 \times 10^{-7} \pm 7.49 \times 10^{-8}$ | $1.48 \times 10^{4}$ | $9.98 \times 10^{-3}$ |
| | CORGI | $3.45 \times 10^{-6} \pm 8.18 \times 10^{-7}$ | $7.50 \times 10^{-6} \pm 9.52 \times 10^{-6}$ | $2.91 \times 10^{-8} \pm 8.68 \times 10^{-9}$ | $1.85 \times 10^{4}$ | $1.22 \times 10^{-2}$ |
| 9 | GNS | $5.67 \times 10^{-6} \pm 1.08 \times 10^{-6}$ | $3.07 \times 10^{-5} \pm 3.60 \times 10^{-5}$ | $1.82 \times 10^{-7} \pm 5.56 \times 10^{-8}$ | $1.58 \times 10^{4}$ | $1.08 \times 10^{-2}$ |
| | CORGI | $2.95 \times 10^{-6} \pm 5.28 \times 10^{-7}$ | $1.47 \times 10^{-5} \pm 1.61 \times 10^{-5}$ | $7.71 \times 10^{-8} \pm 2.00 \times 10^{-8}$ | $1.89 \times 10^{4}$ | $1.24 \times 10^{-2}$ |

Based on the data of Table 10, we actually see that including history hurts performance. We expect this to be the case, since we do not expect second order position derivatives to be very physically meaningful at our timescale. As these do not provide a strong signal, it has the potential to confuse a model. Although the models perform best with a history of 2 (we do not test a history of 1, as this

destroys velocity information), we still report results using a history length of 6 in all of our other experiments, as 6 is the LagrangeBench default, and truncation of the history fundamentally changes the prediction task. We attempt to align with the LagrangeBench default settings so our results are easily compared to others using LagrangeBench.

### E.1.4 INTERPOLATION

Let $D \in \mathbb{N}$ represent the spatial dimension. Then, let us denote the cell size $\Delta_d \in \mathbb{R}$ for each axis $d \in [\![1, N]\!]$. Now, suppose our grid is confined by coordinate bounds $\mathbf{b}_{\min}, \mathbf{b}_{\max} \in \mathbb{R}^D$. Let the feature vector of a particle $p$ be $\mathbf{h}_p \in \mathbb{R}^H$ whose coordinates we denote $\mathbf{i}_p$. We use $\mathbf{G}_{\mathbf{c}} \in \mathbb{R}^H$ to represent the value at the grid cell whose index is the vector $\mathbf{c} \in \mathbb{N}^D$.

For brevity, let us define a normalization function operating on some coordinates

$$\mathbf{u}(\mathbf{i}) = \frac{\mathbf{i} - \mathbf{b}_{\min}}{\mathbf{\Delta}}$$

Then, for each interpolation scheme whose 1-D kernel is denoted by $w(r)$, we use the multidimensional kernel

$$K(\mathbf{i}, \mathbf{c}) = \prod_{d=1}^{D} w\left(\mathbf{u}(\mathbf{i})_d - \mathbf{c}_d - \frac{1}{2}\right)$$

Our scattering operation is thus defined as

$$\mathbf{G}_{\mathbf{c}} = \sum_{p} K(\mathbf{i}_p, \mathbf{c})\mathbf{h}_p$$

and our gathering operation is defined as

$$\mathbf{h}_p = \sum_{\mathbf{c}} K(\mathbf{i}_p, \mathbf{c})\mathbf{G}_{\mathbf{c}}$$

For Nearest Grid Point (Evans & Harlow, 1957), we use the kernel

$$w(r) = \begin{cases} 1, & |r| < \frac{1}{2}, \\ 0, & \text{otherwise.} \end{cases}$$

For Cloud in Cell (Birdsall & Fuss, 1969), we use the kernel

$$w(r) = \max(0, 1 - |r|)$$

For Triangular Shaped Cloud (Eastwood & Hockney, 1974), we use the kernel

$$w(r) = \begin{cases} 0.75 - r^2, & |r| < 0.5, \\ 0.5\,(1.5 - |r|)^2, & 0.5 \leq |r| < 1.5, \\ 0, & |r| \geq 1.5. \end{cases}$$

In Table 11, we find that cloud in cell interpolation works well compared to other options. Surprisingly, using a TSC scatter operation with a NGP gather operation also works well, but we leave investigating this phenomenon further to future work.

Table 11: Performance based on interpolation scheme

| Scatter | Gather | $\text{MSE}_{20}$ | $\text{MSE}_{E_{\text{kin}}}$ | Sinkhorn |
|---------|--------|-------------------|-------------------------------|----------|
| NGP | NGP | $1.90 \times 10^{-6} \pm 3.83 \times 10^{-7}$ | $7.18 \times 10^{-6} \pm 7.76 \times 10^{-6}$ | $6.40 \times 10^{-8} \pm 1.96 \times 10^{-8}$ |
| NGP | CIC | $1.89 \times 10^{-6} \pm 4.02 \times 10^{-7}$ | $6.06 \times 10^{-6} \pm 6.89 \times 10^{-6}$ | $5.31 \times 10^{-8} \pm 1.40 \times 10^{-8}$ |
| NGP | TSC | $1.79 \times 10^{-6} \pm 3.63 \times 10^{-7}$ | $9.80 \times 10^{-6} \pm 1.19 \times 10^{-5}$ | $6.44 \times 10^{-8} \pm 2.01 \times 10^{-8}$ |
| CIC | NGP | $2.10 \times 10^{-6} \pm 4.60 \times 10^{-7}$ | $5.23 \times 10^{-6} \pm 5.48 \times 10^{-6}$ | $4.64 \times 10^{-8} \pm 1.40 \times 10^{-8}$ |
| CIC | CIC | $1.56 \times 10^{-6} \pm 4.27 \times 10^{-7}$ | $4.64 \times 10^{-6} \pm 5.05 \times 10^{-6}$ | $4.45 \times 10^{-8} \pm 1.51 \times 10^{-8}$ |
| CIC | TSC | $1.81 \times 10^{-6} \pm 3.72 \times 10^{-7}$ | $6.52 \times 10^{-6} \pm 7.03 \times 10^{-6}$ | $6.15 \times 10^{-8} \pm 2.20 \times 10^{-8}$ |
| TSC | NGP | $1.55 \times 10^{-6} \pm 4.29 \times 10^{-7}$ | $6.19 \times 10^{-6} \pm 7.44 \times 10^{-6}$ | $2.87 \times 10^{-8} \pm 9.95 \times 10^{-9}$ |
| TSC | CIC | $1.90 \times 10^{-6} \pm 3.56 \times 10^{-7}$ | $7.44 \times 10^{-6} \pm 8.87 \times 10^{-6}$ | $5.27 \times 10^{-8} \pm 1.55 \times 10^{-8}$ |
| TSC | TSC | $1.74 \times 10^{-6} \pm 3.49 \times 10^{-7}$ | $1.09 \times 10^{-5} \pm 1.17 \times 10^{-5}$ | $6.13 \times 10^{-8} \pm 1.86 \times 10^{-8}$ |

## E.2 Tuning

After establishing the empirical soundness of our architecture, we further conduct experiments to probe the effect of various hyperparameters.

### E.2.1 Graph Connectivity

We tested various settings for the graph connectivity radius. Although this is a parameter independent of the model itself, in the sense that this is done more as a data processing step rather than something neural in nature, it significantly affects the performance of our models. We present our findings in terms of the average node degree rather than as the raw connectivity radius so the results may be more interpretable between datasets; the connectivity radius determined in LagrangeBench is derived from the average distance to a neighbor, which is correlated to the average degree inversely.

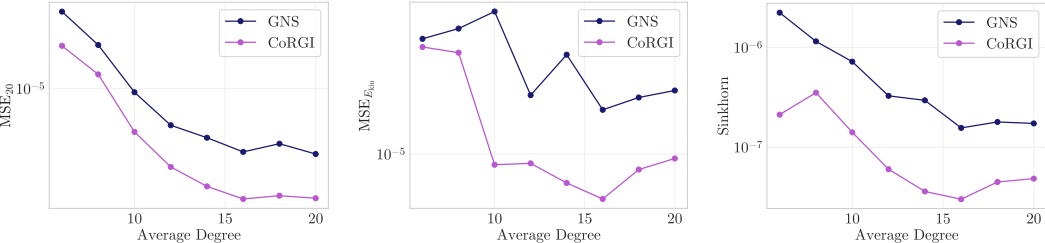

Figure 5: Plot of accuracy compared to average node degree for GNS and CORGI on RPF-2D.

Table 12: Performance based on average vertex degree

| Degree | Model | $\text{MSE}_{20}$ | $\text{MSE}_{E_{\text{kin}}}$ | Sinkhorn | Training time | Inference time |
|--------|-------|-------------------|-------------------------------|----------|---------------|----------------|
| 6 | GNS | $3.64 \times 10^{-5} \pm 4.03 \times 10^{-6}$ | $4.89 \times 10^{-5} \pm 5.41 \times 10^{-5}$ | $2.23 \times 10^{-6} \pm 6.68 \times 10^{-7}$ | $1.43 \times 10^{4}$ | $1.06 \times 10^{-2}$ |
|   | CORGI | $2.05 \times 10^{-5} \pm 1.76 \times 10^{-6}$ | $4.37 \times 10^{-5} \pm 4.34 \times 10^{-5}$ | $2.12 \times 10^{-7} \pm 5.36 \times 10^{-8}$ | $1.90 \times 10^{4}$ | $1.24 \times 10^{-2}$ |
| 8 | GNS | $2.08 \times 10^{-5} \pm 2.81 \times 10^{-6}$ | $5.63 \times 10^{-5} \pm 6.35 \times 10^{-5}$ | $1.15 \times 10^{-6} \pm 2.88 \times 10^{-7}$ | $1.38 \times 10^{4}$ | $9.78 \times 10^{-3}$ |
|   | CORGI | $1.27 \times 10^{-5} \pm 1.41 \times 10^{-6}$ | $4.04 \times 10^{-5} \pm 3.84 \times 10^{-5}$ | $3.52 \times 10^{-7} \pm 9.01 \times 10^{-8}$ | $1.81 \times 10^{4}$ | $1.17 \times 10^{-2}$ |
| 10 | GNS | $9.41 \times 10^{-6} \pm 1.81 \times 10^{-6}$ | $7.12 \times 10^{-5} \pm 7.17 \times 10^{-5}$ | $7.20 \times 10^{-7} \pm 1.90 \times 10^{-7}$ | $1.55 \times 10^{4}$ | $1.10 \times 10^{-2}$ |
|   | CORGI | $4.84 \times 10^{-6} \pm 8.09 \times 10^{-7}$ | $8.64 \times 10^{-6} \pm 9.91 \times 10^{-6}$ | $1.41 \times 10^{-7} \pm 3.97 \times 10^{-8}$ | $1.89 \times 10^{4}$ | $1.22 \times 10^{-2}$ |
| 12 | GNS | $5.41 \times 10^{-6} \pm 1.07 \times 10^{-6}$ | $2.25 \times 10^{-5} \pm 2.44 \times 10^{-5}$ | $3.27 \times 10^{-7} \pm 1.15 \times 10^{-7}$ | $1.50 \times 10^{4}$ | $1.06 \times 10^{-2}$ |
|   | CORGI | $2.69 \times 10^{-6} \pm 5.78 \times 10^{-7}$ | $8.80 \times 10^{-6} \pm 9.05 \times 10^{-6}$ | $6.01 \times 10^{-8} \pm 1.74 \times 10^{-8}$ | $1.72 \times 10^{4}$ | $1.11 \times 10^{-2}$ |
| 14 | GNS | $4.39 \times 10^{-6} \pm 8.48 \times 10^{-7}$ | $3.93 \times 10^{-5} \pm 4.29 \times 10^{-5}$ | $2.95 \times 10^{-7} \pm 1.10 \times 10^{-7}$ | $1.54 \times 10^{4}$ | $1.05 \times 10^{-2}$ |
|   | CORGI | $1.94 \times 10^{-6} \pm 4.45 \times 10^{-7}$ | $6.72 \times 10^{-6} \pm 7.36 \times 10^{-6}$ | $3.60 \times 10^{-8} \pm 1.05 \times 10^{-8}$ | $1.83 \times 10^{4}$ | $1.21 \times 10^{-2}$ |
| 16 | GNS | $3.46 \times 10^{-6} \pm 7.44 \times 10^{-7}$ | $1.84 \times 10^{-5} \pm 1.98 \times 10^{-5}$ | $1.56 \times 10^{-7} \pm 6.04 \times 10^{-8}$ | $1.51 \times 10^{4}$ | $1.00 \times 10^{-2}$ |
|   | CORGI | $1.57 \times 10^{-6} \pm 4.18 \times 10^{-7}$ | $5.39 \times 10^{-6} \pm 6.10 \times 10^{-6}$ | $3.01 \times 10^{-8} \pm 1.03 \times 10^{-8}$ | $1.79 \times 10^{4}$ | $1.15 \times 10^{-2}$ |
| 18 | GNS | $3.97 \times 10^{-6} \pm 7.34 \times 10^{-7}$ | $2.18 \times 10^{-5} \pm 2.37 \times 10^{-5}$ | $1.79 \times 10^{-7} \pm 6.91 \times 10^{-8}$ | $1.57 \times 10^{4}$ | $1.01 \times 10^{-2}$ |
|   | CORGI | $1.66 \times 10^{-6} \pm 3.41 \times 10^{-7}$ | $8.08 \times 10^{-6} \pm 9.15 \times 10^{-6}$ | $4.48 \times 10^{-8} \pm 1.50 \times 10^{-8}$ | $1.76 \times 10^{4}$ | $1.10 \times 10^{-2}$ |
| 20 | GNS | $3.34 \times 10^{-6} \pm 7.09 \times 10^{-7}$ | $2.40 \times 10^{-5} \pm 2.75 \times 10^{-5}$ | $1.73 \times 10^{-7} \pm 7.74 \times 10^{-8}$ | $1.59 \times 10^{4}$ | $1.07 \times 10^{-2}$ |
|   | CORGI | $1.59 \times 10^{-6} \pm 3.09 \times 10^{-7}$ | $9.41 \times 10^{-6} \pm 1.07 \times 10^{-5}$ | $4.84 \times 10^{-8} \pm 1.49 \times 10^{-8}$ | $1.86 \times 10^{4}$ | $1.20 \times 10^{-2}$ |

We see that CORGI consistently outperforms GNS regardless of the average node degree of the input graph. We also see that the performance seems to peak when the graph connectivity is approximately 16, which is similar to the default provided in LagrangeBench. We expect to see that there is no significant difference in training and inference time between the different connectivity settings due to the edge parallelization of GNS, which Table 12 empirically demonstrates. We additionally note that the performance benefit of CORGI persists regardless of the connectivity of the graph.

### E.2.2 CONVOLUTION PARAMETERS

In order to choose the parameters for our global convolution module, we conduct a sweep through various depths and base widths. By depth, we mean the number of different convolutional resolutions involved. For instance, the depiction in Fig 1 has a depth of 3. The test with a depth of 0 serves as an ablation, verifying that our improvements indeed come from genuine convolution operations rather than simply projecting to and from a grid. By width, we refer to the number of channels in the highest resolution layer. At each layer, we double the width and concatenate a skip connection from the graph output of the encoder module.

Table 13: Performance based on convolution depth and width.

| Depth | Width | $\mathrm{MSE}_{20}$ $(\times 10^{-6})$ | $\mathrm{MSE}_{E_{\mathrm{kin}}}$ | Sinkhorn | Training time (s) | Inference time (s) |
|---|---|---|---|---|---|---|
| 0 | – | $3.91 \times 10^{-6} \pm 9.73 \times 10^{-7}$ | $3.29 \times 10^{-5} \pm 4.74 \times 10^{-5}$ | $1.46 \times 10^{-7} \pm 7.49 \times 10^{-8}$ | $9.72 \times 10^{-3}$ | $1.49 \times 10^{4}$ |
| 1 | 32 | $3.18 \times 10^{-6} \pm 7.35 \times 10^{-7}$ | $1.23 \times 10^{-5} \pm 1.52 \times 10^{-5}$ | $1.61 \times 10^{-7} \pm 6.52 \times 10^{-8}$ | $1.58 \times 10^{4}$ | $1.08 \times 10^{-2}$ |
| | 48 | $3.19 \times 10^{-6} \pm 7.21 \times 10^{-7}$ | $1.23 \times 10^{-5} \pm 1.47 \times 10^{-5}$ | $1.83 \times 10^{-7} \pm 7.43 \times 10^{-8}$ | $1.57 \times 10^{4}$ | $1.08 \times 10^{-2}$ |
| | 64 | $3.06 \times 10^{-6} \pm 6.79 \times 10^{-7}$ | $9.87 \times 10^{-6} \pm 1.06 \times 10^{-5}$ | $1.53 \times 10^{-7} \pm 6.76 \times 10^{-8}$ | $1.47 \times 10^{4}$ | $9.89 \times 10^{-3}$ |
| | 96 | $2.96 \times 10^{-6} \pm 6.82 \times 10^{-7}$ | $9.22 \times 10^{-6} \pm 1.05 \times 10^{-5}$ | $1.57 \times 10^{-7} \pm 6.08 \times 10^{-8}$ | $1.56 \times 10^{4}$ | $1.03 \times 10^{-2}$ |
| | 128 | $2.91 \times 10^{-6} \pm 6.46 \times 10^{-7}$ | $9.79 \times 10^{-6} \pm 1.07 \times 10^{-5}$ | $1.78 \times 10^{-7} \pm 8.26 \times 10^{-8}$ | $1.65 \times 10^{4}$ | $1.10 \times 10^{-2}$ |
| | 192 | $3.03 \times 10^{-6} \pm 6.45 \times 10^{-7}$ | $2.10 \times 10^{-5} \pm 2.10 \times 10^{-5}$ | $2.11 \times 10^{-7} \pm 1.12 \times 10^{-7}$ | $1.63 \times 10^{4}$ | $1.12 \times 10^{-2}$ |
| | 256 | $3.14 \times 10^{-6} \pm 7.02 \times 10^{-7}$ | $1.10 \times 10^{-5} \pm 1.20 \times 10^{-5}$ | $1.71 \times 10^{-7} \pm 7.61 \times 10^{-8}$ | $1.62 \times 10^{4}$ | $1.07 \times 10^{-2}$ |
| 2 | 32 | $2.85 \times 10^{-6} \pm 6.39 \times 10^{-7}$ | $1.67 \times 10^{-5} \pm 1.84 \times 10^{-5}$ | $1.49 \times 10^{-7} \pm 6.51 \times 10^{-8}$ | $1.56 \times 10^{4}$ | $1.04 \times 10^{-2}$ |
| | 48 | $3.15 \times 10^{-6} \pm 5.97 \times 10^{-7}$ | $1.32 \times 10^{-5} \pm 1.41 \times 10^{-5}$ | $1.58 \times 10^{-7} \pm 5.26 \times 10^{-8}$ | $1.74 \times 10^{4}$ | $1.16 \times 10^{-2}$ |
| | 64 | $2.68 \times 10^{-6} \pm 5.87 \times 10^{-7}$ | $1.34 \times 10^{-5} \pm 1.49 \times 10^{-5}$ | $1.24 \times 10^{-7} \pm 5.59 \times 10^{-8}$ | $1.65 \times 10^{4}$ | $1.11 \times 10^{-2}$ |
| | 96 | $2.54 \times 10^{-6} \pm 5.73 \times 10^{-7}$ | $1.02 \times 10^{-5} \pm 1.17 \times 10^{-5}$ | $9.39 \times 10^{-8} \pm 3.15 \times 10^{-8}$ | $1.71 \times 10^{4}$ | $1.17 \times 10^{-2}$ |
| | 128 | $2.92 \times 10^{-6} \pm 6.01 \times 10^{-7}$ | $1.06 \times 10^{-5} \pm 1.15 \times 10^{-5}$ | $1.23 \times 10^{-7} \pm 4.57 \times 10^{-8}$ | $1.72 \times 10^{4}$ | $1.18 \times 10^{-2}$ |
| | 192 | $2.40 \times 10^{-6} \pm 5.30 \times 10^{-7}$ | $9.60 \times 10^{-6} \pm 1.14 \times 10^{-5}$ | $9.24 \times 10^{-8} \pm 3.43 \times 10^{-8}$ | $1.65 \times 10^{4}$ | $1.06 \times 10^{-2}$ |
| | 256 | $2.41 \times 10^{-6} \pm 5.79 \times 10^{-7}$ | $9.76 \times 10^{-6} \pm 1.05 \times 10^{-5}$ | $9.71 \times 10^{-8} \pm 3.46 \times 10^{-8}$ | $1.78 \times 10^{4}$ | $1.07 \times 10^{-2}$ |
| 3 | 32 | $1.69 \times 10^{-6} \pm 4.38 \times 10^{-7}$ | $6.97 \times 10^{-6} \pm 7.61 \times 10^{-6}$ | $3.87 \times 10^{-8} \pm 1.54 \times 10^{-8}$ | $1.63 \times 10^{4}$ | $1.08 \times 10^{-2}$ |
| | 48 | $1.75 \times 10^{-6} \pm 4.69 \times 10^{-7}$ | $6.54 \times 10^{-6} \pm 7.00 \times 10^{-6}$ | $4.36 \times 10^{-8} \pm 1.71 \times 10^{-8}$ | $1.67 \times 10^{4}$ | $1.09 \times 10^{-2}$ |
| | 64 | $1.64 \times 10^{-6} \pm 3.50 \times 10^{-7}$ | $6.91 \times 10^{-6} \pm 8.01 \times 10^{-6}$ | $4.52 \times 10^{-8} \pm 1.32 \times 10^{-8}$ | $1.78 \times 10^{4}$ | $1.16 \times 10^{-2}$ |
| | 96 | $1.56 \times 10^{-6} \pm 4.32 \times 10^{-7}$ | $7.14 \times 10^{-6} \pm 8.31 \times 10^{-6}$ | $2.99 \times 10^{-8} \pm 1.11 \times 10^{-8}$ | $1.65 \times 10^{4}$ | $1.07 \times 10^{-2}$ |
| | 128 | $1.53 \times 10^{-6} \pm 4.14 \times 10^{-7}$ | $4.61 \times 10^{-6} \pm 4.80 \times 10^{-6}$ | $2.51 \times 10^{-8} \pm 7.91 \times 10^{-9}$ | $1.75 \times 10^{4}$ | $1.11 \times 10^{-2}$ |
| | 192 | $1.77 \times 10^{-6} \pm 3.77 \times 10^{-7}$ | $8.21 \times 10^{-6} \pm 9.21 \times 10^{-6}$ | $8.13 \times 10^{-8} \pm 1.97 \times 10^{-8}$ | $1.94 \times 10^{4}$ | $1.18 \times 10^{-2}$ |
| | 256 | $1.69 \times 10^{-6} \pm 3.50 \times 10^{-7}$ | $6.14 \times 10^{-6} \pm 7.45 \times 10^{-6}$ | $4.49 \times 10^{-8} \pm 1.33 \times 10^{-8}$ | $2.15 \times 10^{4}$ | $1.20 \times 10^{-2}$ |
| 4 | 32 | $1.69 \times 10^{-6} \pm 4.26 \times 10^{-7}$ | $6.07 \times 10^{-6} \pm 6.75 \times 10^{-6}$ | $3.07 \times 10^{-8} \pm 7.75 \times 10^{-9}$ | $1.88 \times 10^{4}$ | $1.26 \times 10^{-2}$ |
| | 48 | $1.59 \times 10^{-6} \pm 4.09 \times 10^{-7}$ | $4.92 \times 10^{-6} \pm 5.86 \times 10^{-6}$ | $2.28 \times 10^{-8} \pm 5.88 \times 10^{-9}$ | $1.89 \times 10^{4}$ | $1.20 \times 10^{-2}$ |
| | 64 | $1.88 \times 10^{-6} \pm 3.87 \times 10^{-7}$ | $7.42 \times 10^{-6} \pm 8.50 \times 10^{-6}$ | $4.53 \times 10^{-8} \pm 1.02 \times 10^{-8}$ | $1.90 \times 10^{4}$ | $1.25 \times 10^{-2}$ |
| | 96 | $1.77 \times 10^{-6} \pm 3.58 \times 10^{-7}$ | $1.22 \times 10^{-5} \pm 1.40 \times 10^{-5}$ | $3.60 \times 10^{-8} \pm 8.79 \times 10^{-9}$ | $1.95 \times 10^{4}$ | $1.25 \times 10^{-2}$ |
| | 128 | $1.56 \times 10^{-6} \pm 4.08 \times 10^{-7}$ | $4.17 \times 10^{-6} \pm 4.37 \times 10^{-6}$ | $2.20 \times 10^{-8} \pm 5.40 \times 10^{-9}$ | $2.06 \times 10^{4}$ | $1.20 \times 10^{-2}$ |
| | 192 | $1.52 \times 10^{-6} \pm 4.02 \times 10^{-7}$ | $4.31 \times 10^{-6} \pm 4.83 \times 10^{-6}$ | $1.72 \times 10^{-8} \pm 4.94 \times 10^{-9}$ | $2.52 \times 10^{4}$ | $1.25 \times 10^{-2}$ |
| | 256 | $1.67 \times 10^{-6} \pm 3.55 \times 10^{-7}$ | $5.89 \times 10^{-6} \pm 6.66 \times 10^{-6}$ | $3.70 \times 10^{-8} \pm 1.03 \times 10^{-8}$ | $3.10 \times 10^{4}$ | $1.32 \times 10^{-2}$ |

Matching our analysis in App A, we find that a depth of 3 or 4 is optimal, as we obtain a roughly global receptive field given our initial resolution on RPF-2D of $32 \times 64$.

We moreover show that increasing our width has diminishing returns, and in particular, beyond a width of 128 on depths of 3 and 4 we actually see a degradation in performance. Since we utilized a backbone of GNS with a latent dimension of 128, we believe that the choice of width in our global convolution module should simply be inherited from the graph encoder module.

We note that increasing the width approximately maintains the inference time, but generally increases the training time. Predictably, increasing the depth will increase both the inference time and the training time.

We also hypothesize that the density of our grid has a significant impact on performance. We attempt to justify empirically that our choice (i.e. a density of one or two particles per voxel) in Table 14.

Table 14: Performance based on convolution resolution

| Resolution | $\text{MSE}_{20}$ | $\text{MSE}_{E_{\text{kin}}}$ | Sinkhorn | Training time | Inference time |
|---|---|---|---|---|---|
| $16 \times 32$ | $1.78 \times 10^{-6} \pm 3.58 \times 10^{-7}$ | $6.17 \times 10^{-6} \pm 7.23 \times 10^{-6}$ | $3.43 \times 10^{-8} \pm 8.05 \times 10^{-9}$ | $1.72 \times 10^{4}$ | $1.08 \times 10^{-2}$ |
| $32 \times 64$ | $1.54 \times 10^{-6} \pm 4.17 \times 10^{-7}$ | $4.58 \times 10^{-6} \pm 5.33 \times 10^{-6}$ | $2.37 \times 10^{-8} \pm 7.71 \times 10^{-9}$ | $1.72 \times 10^{4}$ | $1.09 \times 10^{-2}$ |
| $48 \times 96$ | $2.57 \times 10^{-6} \pm 4.71 \times 10^{-7}$ | $1.25 \times 10^{-5} \pm 1.43 \times 10^{-5}$ | $1.59 \times 10^{-7} \pm 5.60 \times 10^{-8}$ | $1.78 \times 10^{4}$ | $1.04 \times 10^{-2}$ |
| $64 \times 128$ | $2.49 \times 10^{-6} \pm 5.77 \times 10^{-7}$ | $1.33 \times 10^{-5} \pm 1.53 \times 10^{-5}$ | $1.09 \times 10^{-7} \pm 4.69 \times 10^{-8}$ | $1.94 \times 10^{4}$ | $1.07 \times 10^{-2}$ |

The data show that our choice of $32 \times 64$ performs well given our other default settings and our chosen aspect ratio. The drastic decrease in performance in higher resolutions compared to the relatively minor decrease in performance in the lower resolution corroborates our claim that we need for nodes to have a global receptive field: higher resolutions require more than 3 layers of global convolution, but in Table 14, we test using only 3 layers.

### E.2.3 LEARNING PARAMETERS

We further assess the impact of CORGI on learning parameters.

Table 15: Performance based on learning rate and batch size

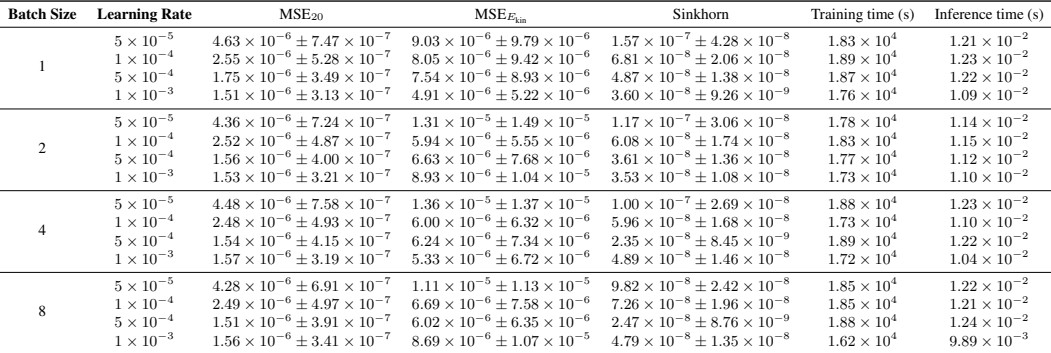

| Batch Size | Learning Rate | $\text{MSE}_{20}$ | $\text{MSE}_{E_{\text{kin}}}$ | Sinkhorn | Training time (s) | Inference time (s) |
|---|---|---|---|---|---|---|
| 1 | $5 \times 10^{-5}$ | $4.63 \times 10^{-6} \pm 7.47 \times 10^{-7}$ | $9.03 \times 10^{-6} \pm 9.79 \times 10^{-6}$ | $1.57 \times 10^{-7} \pm 4.28 \times 10^{-8}$ | $1.83 \times 10^{4}$ | $1.21 \times 10^{-2}$ |
| | $1 \times 10^{-4}$ | $2.55 \times 10^{-6} \pm 5.28 \times 10^{-7}$ | $8.05 \times 10^{-6} \pm 9.42 \times 10^{-6}$ | $6.81 \times 10^{-8} \pm 2.06 \times 10^{-8}$ | $1.89 \times 10^{4}$ | $1.23 \times 10^{-2}$ |
| | $5 \times 10^{-4}$ | $1.75 \times 10^{-6} \pm 3.49 \times 10^{-7}$ | $7.54 \times 10^{-6} \pm 8.93 \times 10^{-6}$ | $4.87 \times 10^{-8} \pm 1.38 \times 10^{-8}$ | $1.87 \times 10^{4}$ | $1.22 \times 10^{-2}$ |
| | $1 \times 10^{-3}$ | $1.51 \times 10^{-6} \pm 3.13 \times 10^{-7}$ | $4.91 \times 10^{-6} \pm 5.22 \times 10^{-6}$ | $3.60 \times 10^{-8} \pm 9.26 \times 10^{-9}$ | $1.76 \times 10^{4}$ | $1.09 \times 10^{-2}$ |
| 2 | $5 \times 10^{-5}$ | $4.36 \times 10^{-6} \pm 7.24 \times 10^{-7}$ | $1.31 \times 10^{-5} \pm 1.49 \times 10^{-5}$ | $1.17 \times 10^{-7} \pm 3.06 \times 10^{-8}$ | $1.78 \times 10^{4}$ | $1.14 \times 10^{-2}$ |
| | $1 \times 10^{-4}$ | $2.52 \times 10^{-6} \pm 4.87 \times 10^{-7}$ | $5.94 \times 10^{-6} \pm 5.55 \times 10^{-6}$ | $6.08 \times 10^{-8} \pm 1.74 \times 10^{-8}$ | $1.83 \times 10^{4}$ | $1.15 \times 10^{-2}$ |
| | $5 \times 10^{-4}$ | $1.56 \times 10^{-6} \pm 4.00 \times 10^{-7}$ | $6.63 \times 10^{-6} \pm 7.68 \times 10^{-6}$ | $3.61 \times 10^{-8} \pm 1.36 \times 10^{-8}$ | $1.77 \times 10^{4}$ | $1.12 \times 10^{-2}$ |
| | $1 \times 10^{-3}$ | $1.53 \times 10^{-6} \pm 3.21 \times 10^{-7}$ | $8.93 \times 10^{-6} \pm 1.04 \times 10^{-5}$ | $3.53 \times 10^{-8} \pm 1.08 \times 10^{-8}$ | $1.73 \times 10^{4}$ | $1.10 \times 10^{-2}$ |
| 4 | $5 \times 10^{-5}$ | $4.48 \times 10^{-6} \pm 7.58 \times 10^{-7}$ | $1.36 \times 10^{-5} \pm 1.37 \times 10^{-5}$ | $1.00 \times 10^{-7} \pm 2.69 \times 10^{-8}$ | $1.88 \times 10^{4}$ | $1.23 \times 10^{-2}$ |
| | $1 \times 10^{-4}$ | $2.48 \times 10^{-6} \pm 4.93 \times 10^{-7}$ | $6.00 \times 10^{-6} \pm 6.32 \times 10^{-6}$ | $5.96 \times 10^{-8} \pm 1.68 \times 10^{-8}$ | $1.73 \times 10^{4}$ | $1.10 \times 10^{-2}$ |
| | $5 \times 10^{-4}$ | $1.54 \times 10^{-6} \pm 4.15 \times 10^{-7}$ | $6.24 \times 10^{-6} \pm 7.34 \times 10^{-6}$ | $2.35 \times 10^{-8} \pm 8.45 \times 10^{-9}$ | $1.89 \times 10^{4}$ | $1.22 \times 10^{-2}$ |
| | $1 \times 10^{-3}$ | $1.57 \times 10^{-6} \pm 3.19 \times 10^{-7}$ | $5.33 \times 10^{-6} \pm 6.72 \times 10^{-6}$ | $4.89 \times 10^{-8} \pm 1.46 \times 10^{-8}$ | $1.72 \times 10^{4}$ | $1.04 \times 10^{-2}$ |
| 8 | $5 \times 10^{-5}$ | $4.28 \times 10^{-6} \pm 6.91 \times 10^{-7}$ | $1.11 \times 10^{-5} \pm 1.13 \times 10^{-5}$ | $9.82 \times 10^{-8} \pm 2.42 \times 10^{-8}$ | $1.85 \times 10^{4}$ | $1.22 \times 10^{-2}$ |
| | $1 \times 10^{-4}$ | $2.49 \times 10^{-6} \pm 4.97 \times 10^{-7}$ | $6.69 \times 10^{-6} \pm 7.58 \times 10^{-6}$ | $7.26 \times 10^{-8} \pm 1.96 \times 10^{-8}$ | $1.85 \times 10^{4}$ | $1.21 \times 10^{-2}$ |
| | $5 \times 10^{-4}$ | $1.51 \times 10^{-6} \pm 3.91 \times 10^{-7}$ | $6.02 \times 10^{-6} \pm 6.35 \times 10^{-6}$ | $2.47 \times 10^{-8} \pm 8.76 \times 10^{-9}$ | $1.88 \times 10^{4}$ | $1.24 \times 10^{-2}$ |
| | $1 \times 10^{-3}$ | $1.56 \times 10^{-6} \pm 3.41 \times 10^{-7}$ | $8.69 \times 10^{-6} \pm 1.07 \times 10^{-5}$ | $4.79 \times 10^{-8} \pm 1.35 \times 10^{-8}$ | $1.62 \times 10^{4}$ | $9.89 \times 10^{-3}$ |

In Table 15, we find empirically that our optimal learning rates match that of LagrangeBench. We expect that CORGI will not require a significant change in learning parameters, reducing much experimental overhead in incorporating CORGI in real world prediction tasks.

### E.3 LONGER TIMES

In this section, we qualitatively evaluate our performance over longer time frames in the DAM-2D dataset. We choose this dataset as its high turbulence represents a more realistic usecase and is more succeptible to the butterfly effect, hence making long term predictions more challenging.

We compare our performance to those found in the Neural SPH paper (Toshev et al., 2024), but are unable to reproduce their results: despite multiple attempted revisions, we could not successfully run the code provided by Toshev et al. Their code includes references to modules not included in the camera ready version, and hence, is not executable in its released form. As we could not reproduce their experiments, we simply restate their results and compare them against our own. They did not report significance, hence it is not present in our table. We borrow their notation, where $\square_g$ has external forces removed and $\square_p$ includes a pressure term.

Table 16: Performance compared to Neural SPH.

| Model | $\text{MSE}_{400}$ | $\text{MSE}_{E_{\text{kin}}}$ | Sinkhorn |
|---|---|---|---|
| GNS | $1.05 \times 10^{-1} \pm 2.05 \times 10^{-2}$ | $2.22 \times 10^{-2} \pm 1.24 \times 10^{-2}$ | $1.91 \times 10^{-2} \pm 9.57 \times 10^{-3}$ |
| $\text{GNS}_g$ | $8.0 \times 10^{-2}$ | $1.3 \times 10^{-2}$ | $9.4 \times 10^{-3}$ |
| $\text{GNS}_p$ | $9.7 \times 10^{-2}$ | $7.1 \times 10^{-3}$ | $5.8 \times 10^{-3}$ |
| $\text{GNS}_{g,p}$ | $8.4 \times 10^{-2}$ | $7.5 \times 10^{-3}$ | $2.1 \times 10^{-3}$ |
| SEGNN | $1.71 \times 10^{-1} \pm 1.01 \times 10^{-2}$ | $1.21 \times 10^{-2} \pm 1.83 \times 10^{-3}$ | $3.09 \times 10^{-2} \pm 5.98 \times 10^{-3}$ |
| $\text{SEGNN}_g$ | $1.6 \times 10^{-1}$ | $2.1 \times 10^{-2}$ | $1.9 \times 10^{1}$ |
| $\text{SEGNN}_p$ | $1.2 \times 10^{-1}$ | $9.4 \times 10^{-3}$ | $5.2 \times 10^{-2}$ |
| $\text{SEGNN}_{g,p}$ | $8.6 \times 10^{-2}$ | $4.9 \times 10^{-3}$ | $2.6 \times 10^{-3}$ |
| CoRGI | $2.84 \times 10^{-2} \pm 2.53 \times 10^{-3}$ | $3.04 \times 10^{-4} \pm 8.79 \times 10^{-5}$ | $9.59 \times 10^{-4} \pm 1.96 \times 10^{-4}$ |

From our experiments, we conclude that CoRGI performs substantially better on the DAM-2D dataset. However, it is possible to use Neural SPH with CoRGI in theory, and we can potentially see even better performance. We additionally provide qualitative analysis of how well CoRGI performs with longer rollouts.

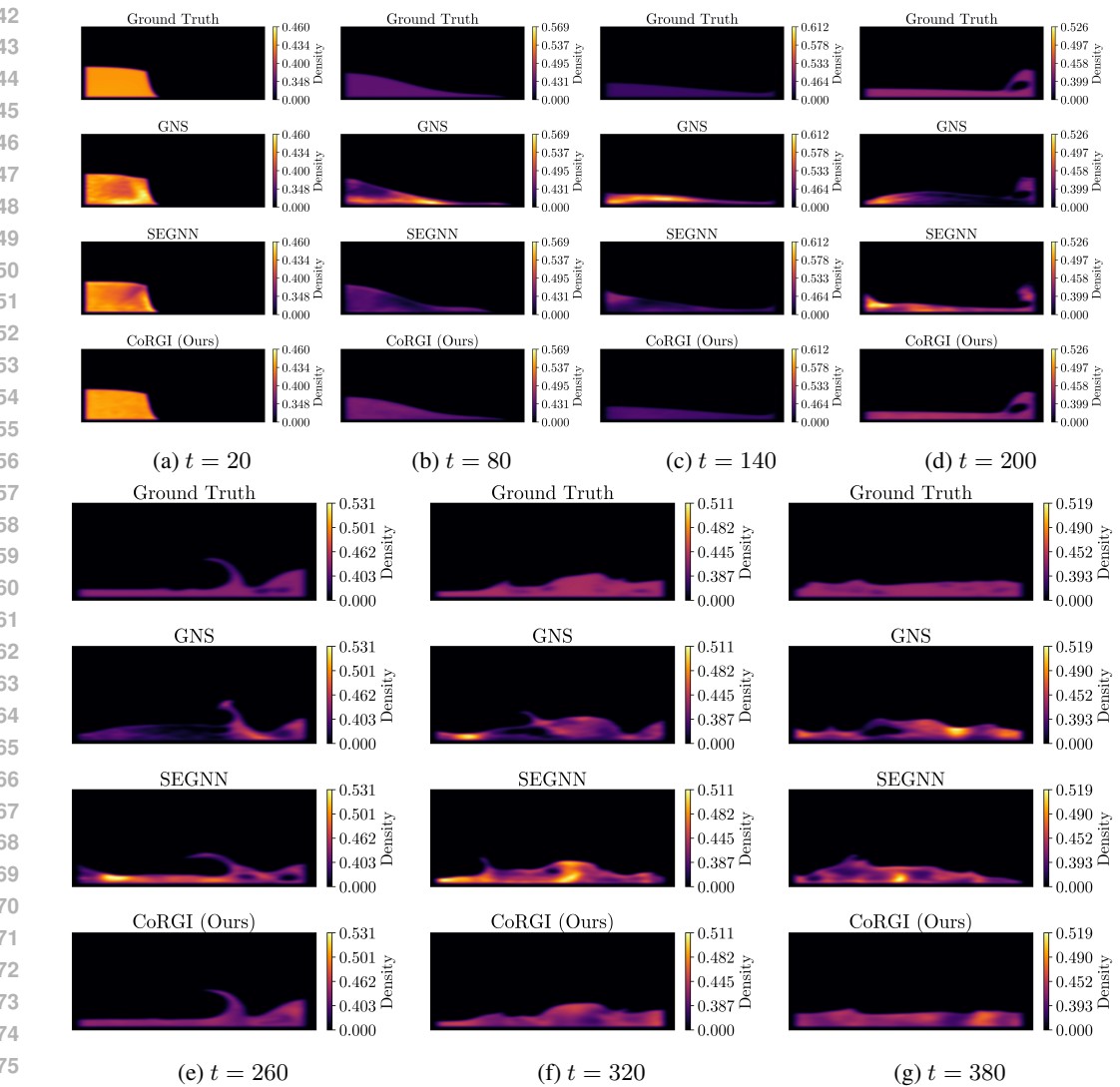

Figure 6: Kernel density estimation on DAM-2D ($t \in [\![0, 400]\!]$) on a single rollout. Above, uniform coloring indicates adherence to fluid incompressibility. Qualitatively, CORGI is able to remain close to the ground truth for the entire rollout.

As established in Fig 2, even in short time frames, both GNS and SEGNN struggle to maintain fluid incompressibility, and thus, failed to learn robust physical patterns. We see that GNS and SEGNN introduce density issues even as early as $t = 20$. The density issues on GNS are especially apparent in the images for $t = 80$ and $t = 140$, and still remain prominent on SEGNN. By the time the fluid has hit the wall at the other side and formed a wave, we see CORGI still maintains a relatively close distribution to the ground truth, whereas both GNS and SEGNN continue to struggle. Whereas before the GNS and SEGNN still predicted a similar silhouette, we see that at around $t = 200$ the shapes themselves become distorted. This is not a problem for CORGI even up to $t = 380$. However, after $t = 200$, our model's density predictions do become slightly inconsistent with the ground truth, and we hypothesize that SPH relaxation may be used here to address it.

Our results on the other dataset in Table 17 show that in less turbulent cases, the performance of CORGI is still comparable to NeuralSPH on long rollouts.

Table 17: Performance for MSE400 as defined by Toshev et al. (2024). As we were unable to reproduce the results of NeuralSPH, we utilize their results for GNS and NeuralSPH.

| Dataset | Model | $\mathrm{MSE}_{400}$ | $\mathrm{MSE}_{E_{\mathrm{kin}}}$ | Sinkhorn |
|---------|-------|------|------|----------|
| DAM-2D | GNS | $1.9 \times 10^{-1}$ | $4.6 \times 10^{-2}$ | $3.8 \times 10^{-2}$ |
| | NeuralSPH | $8.4 \times 10^{-2}$ | $2.1 \times 10^{-3}$ | $7.5 \times 10^{-3}$ |
| | CoRGI | $2.84 \times 10^{-2}$ | $3.04 \times 10^{-4}$ | $9.59 \times 10^{-4}$ |
| LDC-2D | GNS | $3.3 \times 10^{-2}$ | $1.1 \times 10^{-4}$ | $3.1 \times 10^{-4}$ |
| | NeuralSPH | $1.6 \times 10^{-2}$ | $1.2 \times 10^{-6}$ | $2.8 \times 10^{-7}$ |
| | CoRGI | $1.64 \times 10^{-2}$ | $1.44 \times 10^{-4}$ | $5.00 \times 10^{-5}$ |
| RPF-2D | GNS | $2.7 \times 10^{-2}$ | $4.3 \times 10^{-3}$ | $3.7 \times 10^{-7}$ |
| | NeuralSPH | $2.7 \times 10^{-2}$ | $1.4 \times 10^{-4}$ | $3.0 \times 10^{-8}$ |
| | CoRGI | $2.28 \times 10^{-2}$ | $2.60 \times 10^{-4}$ | $3.93 \times 10^{-7}$ |
| TGV-2D | GNS | $5.3 \times 10^{-4}$ | $5.6 \times 10^{-7}$ | $5.4 \times 10^{-7}$ |
| | NeuralSPH | $4.8 \times 10^{-4}$ | $4.8 \times 10^{-7}$ | $1.7 \times 10^{-8}$ |
| | CoRGI | $4.55 \times 10^{-4}$ | $4.22 \times 10^{-7}$ | $4.78 \times 10^{-7}$ |
| LDC-3D | GNS | $3.2 \times 10^{-2}$ | $1.3 \times 10^{-7}$ | $2.0 \times 10^{-5}$ |
| | NeuralSPH | $3.2 \times 10^{-2}$ | $2.9 \times 10^{-8}$ | $1.1 \times 10^{-6}$ |
| | CoRGI | $3.23 \times 10^{-2}$ | $3.12 \times 10^{-8}$ | $1.04 \times 10^{-6}$ |

### E.4 UPT

As the UPT authors provide configurations to obtain MSE20 over the LagrangeBench datasets, we provide a comparison in Table 18.

Table 18: Performance for MSE20 against UPT.

| Model | DAM-2D | LDC-2D | RPF-2D | TGV-2D | RPF-3D | TGV-3D |
|-------|--------|--------|--------|--------|--------|--------|
| UPT | $4.28 \times 10^{-4}$ | $3.34 \times 10^{-3}$ | $1.12 \times 10^{-2}$ | $1.11 \times 10^{-2}$ | $5.33 \times 10^{-4}$ | $1.06$ |
| CoRGI | $1.55 \times 10^{-5}$ | $1.45 \times 10^{-5}$ | $1.54 \times 10^{-6}$ | $3.81 \times 10^{-6}$ | $1.95 \times 10^{-5}$ | $6.10 \times 10^{-3}$ |

We believe that this difference is due to the different objectives of the models. UPT is designed to model the velocity field directly, and hence is significantly weaker than CoRGI at predicting particle trajectories across all datasets. Hence, we do not find the models to be directly comparable, although CoRGI may be used for UPT's applications and UPT may be used for CoRGI's applications. In terms of inference time, the two methods are comparable: e.g., CoRGI averages 0.013 seconds per rollout step on TGV-2D, while UPT averages 0.014 seconds.

## F LIMITATIONS AND FUTURE WORK

**Resolution Coupling.** Our method's fixed Eulerian grid can oversmooth dense particle regions and inefficiently allocate resolution to sparse areas, acting as a low-pass filter. This uniform spacing struggles with non-uniform particle distributions, potentially losing sub-grid details. Adaptive grids (e.g., octrees, deformable kernels) could help, but maintaining the simplicity and efficiency of regular convolutions is an open challenge.

**Grid-Induced Anisotropy.** The axis-aligned grid introduces orientation bias, potentially representing features aligned with grid axes more sharply than others (e.g., diagonal shocks may "stairstep"). Potential solutions include running multiple CNN branches with rotated projections or using group-equivariant/steerable convolutions for built-in rotational equivariance, with the trade-off being increased computational cost.

## LLM USAGE

LLMs were used to polish writing and to help debug code.

