# OpenReview forum: "CoRGI: GNNs with Convolutional Residual Global Interaction for Lagrangian Simulation"
_ICLR.cc/2026/Conference — ICLR 2026 Conference Withdrawn Submission_

### Official Review · Reviewer_CtMQ · 2025-10-28

**Soundness:** 3
**Presentation:** 4
**Contribution:** 1
**Rating:** 2
**Confidence:** 5

**Summary:**

The paper addresses a key challenge - How to handle global fluid information in highly dynamic fluid flows. they propose a hybrid Euler-Lagrange scheme, wherein they leverage a residual CNN layer to perform global convolution. With extensive experiments, they show that the architecture outperforms GNS.

**Strengths:**

The motivation for using global feature processing seems relevant in context to capturing dynamic fluid flows.
The writing is very good. All the components are clearly explained.
Experiments are extensive.

**Weaknesses:**

Convolution is not discretization agnostic, whereas GNN is discretization invariant. Unlike standard global convolution module, (i.e. FNO, GNOT, UPT, Transolver, FigConvNet CODANO), this approach seems to be applying "Local convolution" repeatedly for information aggregation. This is almost equivalent to applying GNN layers repeatedly, as the information aggregation would eventually propagate to the ends of the graph (oversmoothing). Can the authors show how many GNN layers are required to achieve the same performance as using U-Net autoencoder? GNS seems to be faster, but it is not clear if increasing the number of layers/training steps would improve GNS accuracy.

The notion of projecting to a grid to propagate is not new. Several models already do that (GINO, GIOROM, UPT, Transolver) and these models are discretization agnostic, and enable latent feature learning through global kernel integral transforms and fourier neural operators. The difference here, however, is that this is used as an aggregator, rather than as a latent-space propagation module.

Additionally, combining eulerian and Lagrangian schemes is not new - "Hybrid Neural-MPM for Interactive Fluid Simulations
in Real-Time, Jingxuan Xu"

"A Neural Material Point Method for Particle-based Emulation, Omer Rochman-Sharabi"

**Questions:**

My primary question is the notion of using local convolutions repeatedly, as opposed to using GNN layers repeatedly. It seems like in Table 1, GNS is faster, so I fail to see the advantage of CNN, unless the addition of more GNN layers would slow down GNS, but how would that affect the accuracy of GNS? Could the authors justify how those two are not equivalent?

Moreover, many latent space methods project to regular grids. The reduction aspect of U-Net, serves as a form of model-order reduction. But in this case, it's not clear if the reduced latents are used to timestep, or simply aggregate and lift back to original space. Wouldn't multi-pole graph kernel network serve the same purpose (V-cycle aggregation)?

Why have the authors not considered global convolutions for global feature learning? (i.e GNOT, UPT, Transolver, GIOROM, GINO, FNO, CoDANO, multipole graph kernel operator, deeponet etc.). UPT does not explicitly model Lagrangian datasets, as it cannot model autoregression, but all of the other models are capable of performing global convolutions, so why use local convolutions (U-NET)?

---

### Official Review · Reviewer_wExu · 2025-10-30

**Soundness:** 3
**Presentation:** 3
**Contribution:** 2
**Rating:** 2
**Confidence:** 4

**Summary:**

CoRGI is an extension to message passing networks that allows global interactions.
It is motivated by hybrid numerical methods such as particle in cell and FLIP, which use both eulerian and lagrangian in a particle-mesh solver.

The authors validate their model on LagrangeBench, a particle-based and GNN-focused benchmark for hydrodynamics.
CoRGI can be applied on top of existing models, such as GNS, achieving a relatively cheap ~50% performance improvement.

**Strengths:**

- Writing is generally easy to understand and follow, the figurework is also helpful and well designed. Motivation for the method is clearly stated and supported by "legacy" choices in traditional numerics (eg PIC).
- Results are impressive, with only marginal added inference cost CoRGI significantly improves long term performance. Result presentation could be improved, as table 1 is rather large and not highlighted. Figure 3 is also interesting, but could be displayed better, as some metrics are semilog-y and others are linear. Generally I find that semilog plots are hard to read.
- Interpretable results: sinkhorn is a distribution distance, and the significant improvement if it reflects the explicit global nature of the method.
- The method is understandable and not overstated, section 3.2 is welcome and fits well to support the method. I did not check the correctness of the proof.
- Extensive ablations in the appendix on interpolation schemes, graph and CNN architectures.

**Weaknesses:**

My main concern with the paper can be summarized with a mismatch between motivation, model architecture and experiments supporting the claims. While developed in the realm of neural surrogates, it also fails to account for very similar graph neural network research. For this reason I don't think the paper is ready to be accepted. If the authors can provide preliminary evaluations on larger systems, and especially systems with long range / global interactions, I will happily raise my score.

### Dataset choice
Despite proposing a local-global method, the authors only verify it on a lagrangian dataset generated with SPH, an local-only (particle) method. It represents fluids as particles, and field quantities are computed through (smoothing) kernels over radial neighborhoods, with the assumption of small enough timesteps. It works very well for local variations and free surfaces, but it is not explicitly global. Admittedly lagrangebench datasets are time-coarsened with a rate of 100x, breaking the strict locality of SPH.

Regardless, it would be very beneficial to velidate CoRGI on different settings, namely purely eulerian datasets (e.g. turbulent PDEs on grids) and graph datasets with explicit global/long range interactions (e.g. molecular dynamics like rMD17, generic graph benchmark such as LRGB https://arxiv.org/abs/2309.00367).
Finally, data generated with hybrid solverslike PIC would be the ideal setup, although I am not aware of benchmarks on the latter.

### Model comparisons
Comparison to other message passing + global interaction techniques is missing, for instance Ewald-MP (https://arxiv.org/abs/2303.04791) or MFN (https://arxiv.org/abs/2310.10434).

Due to the similarity of application areas, a combination with NeuralSPH would also be interesting, as mentioned in the related works (line 140).

I do not entirely agree with lines 454-456: _While MSE20 and MSEEkin are primarily local errors [...], Sinkhorn divergence [...] is global in nature._. It is true that Sinkhorn is global in nature, but because of its 2000 timestep horizon, MSE20 while pointwise also contains some long range, non-local information about the interaction of particles.
My interpretation is that on long time windows an autoregressive neural surrogate will inevitably accumulate errors and particle pairings will become meaningless;
in turn, Sinkhorn does not have node pairs and looks at the distribution, which is meaningful even with mismatched nodes. MSE can be very high if the individual particles are relocated, even if the predictions are stable.

### Writing
The introduction could be restructured to be more concise: for instance, lines 047-049 are reworded and repeated in line 071-072. Also there are some minor inaccuracies, for example line 064 "Despite their expressiveness, GNNs are computationally limited". The word expressiveness is misused here, as on one hand it makes the sentence sound like a contradiction, and on the other message passing is strictly upper bounded by the 1-WL test, making it non-expressive by definition.

A more in-depth result discussion would be beneficial: for example, why is dam break seemingly the best performing dataset for CoRGI?

### Scalability
LagrangeBench only goes up to ~10K particles, as GNNs are not trivial to scale to significantly larger systems due to computational (memory) requirements. Similarly, CNNs can be applied to large grids, but ViTs mostly overtook convolution in performance scalability on large grids and large amount of data (e.g. in scientific ML https://arxiv.org/abs/2411.09678, https://arxiv.org/abs/2502.02414).


### Limitations
Limitations are at the end of the appendix, and I think it's important to have them in the main body. Regardless, the first aspect (resolution coupling) is extremely relevant in application: all datasets in lagrangebench except DAM are fully packed grids with SPH-relaxed particle distribution, meaning that this weakness is not explored in this work. I agree that an adaptive / sparse approach is a valuable future work.

**Questions:**

- the most impressive promotion is seen on dam break. do you have an interpretation as to why?
- what do the authors mean by "which we address in CORGI by increasing the receptive field of each node" (line 129)? to me this sounds like the radius of the graph was increased for CoRGI, was the same done for GNS/SEGNN?
- Is there any investigation conducted in the direction of scalability?
- SPH is inherently a local method. I understand why machine learning requires such long horizons in a framework such as lagrangebench (time coarsening -> large interaction radius), but why is in general a global method required for modeling SPH simulations?

---

### Official Review · Reviewer_2nk4 · 2025-10-31

**Soundness:** 2
**Presentation:** 2
**Contribution:** 2
**Rating:** 4
**Confidence:** 4

**Summary:**

This paper aims to address the challenge of modeling long-range interactions in particle-based fluid simulation. To achieve this, the authors propose a hybrid framework that combines particle and grid representations: particle features are projected onto a grid for convolutional updates and subsequently mapped back to the particle domain. Experiments conducted on several Lagrangian benchmark datasets demonstrate improvements over existing methods.

**Strengths:**

* The proposed method achieves superior performance compared to existing baselines.
* It maintains computational efficiency, introducing only minimal additional overhead.

**Weaknesses:**

1. Limited evaluation horizon. Results are reported only for 20-step rollouts, which is atypical for practical use. This short horizon makes it difficult to assess **error accumulation and long-term stability**. Please report long-term prediction performance (e.g., full-trajectory rollouts of $\ge$ 150 frames, as in GNS).
2. Insufficient temporal qualitative evidence. The qualitative evaluation relies on sparsely sampled frames, which obscures temporal coherence (smoothness, stability, and plausibility of motion). Side-by-side videos of predictions vs. ground truth at fixed frame rates would better demonstrate dynamics, including representative failure cases.
3. Missing 3D visualizations for 3D tasks. Although the experiments include 3D scenarios, only 2D snapshots are shown. Please provide 3D renderings for the 3D cases.

**Questions:**

Please refer to the weaknesses.

**Details Of Ethics Concerns:**

My primary concern is the limited evaluation, which does not sufficiently demonstrate the method’s effectiveness. See the weaknesses for details.

---

### Official Review · Reviewer_nrPq · 2025-10-31

**Soundness:** 2
**Presentation:** 3
**Contribution:** 2
**Rating:** 4
**Confidence:** 4

**Summary:**

The paper proposed CoRGI, a hybrid GNN-CNN simulators: particle features are scattered to a grid, processed by a multi-resolution CNN/U-Net, then gathered back to particles and decoded by the GNN, which captures long-range/global interactions with modest overhead. On LagrangeBench, CORGI improves rollout accuracy versus GNS/SEGNN while remaining efficient in training/inference time.

**Strengths:**

1. The paper is clear written and the design choice is well-motivated.

2. The proposed architecture balanced accuracy–time trade-off quite well. Gains in rollout error come with modest runtime/memory overhead, under matched budgets it can outperform pure GNN baselines.

3. By using CFL condition to determine a effective spatial length scale, the paper argues that too few message passing steps can be detrimental to the performance and shows multiscale CNN/U-Net layers achieve much faster global information mixing than GNN message passing at fixed depth.

**Weaknesses:**

The core idea: projecting unstructured grid to a uniform Eulerian grid to exploit efficient convolutions/spectral operaions—has been explored (e.g., [1] uses graph kernel network to project input onto a equi-spaced grid for effcient Fourier transform), and U-shaped/UNet-style GNN designs for simulation are also established (e.g., [2]). In addition to CNN/GNN hybrid architecture or U-shape architecture, there is also Transformer-based architecture like [3] that can handle Eulerian/Lagrangian simulation with global interaction captured efficiently.

As a result, the contribution currently feels incremental, and the specific conceptual advance isn’t fully clear. It would be helpful to more explicitly differentiate from prior works.

[1] Li, Zongyi, et al. "Geometry-informed neural operator for large-scale 3d pdes."

[2] Cao, Yadi, et al. "Efficient learning of mesh-based physical simulation with bsms-gnn."

[3] Alkin, Benedikt, et al. "Universal physics transformers: A framework for efficiently scaling neural operators."

**Questions:**

What happens under large deformations or domain growth/shrinkage, is there support for adaptive grid resolution? Like re-mesh/re-parameterize the uniform grid each step. This is quite important, since Lagrangian simulations are typically used in settings with large deformations and moving boundaries

---

### Note · Authors · 2025-11-12

I have read and agree with the venue's withdrawal policy on behalf of myself and my co-authors.